# Evidence for EPP and QBO modulations of the Antarctic $NO_2$ springtime stratospheric column from OMI observations

Emily Gordon[1], Annika Seppälä[1], and Johanna Tamminen[2]

[1]Department of Physics, University of Otago, Dunedin, New Zealand
[2]Space and Earth Observation Centre, Finnish Meteorological Institute, Helsinki, Finland

**Correspondence:** Annika Seppälä (annika.seppala@otago.ac.nz)

**Abstract.** Observations from the Ozone Monitoring Instrument (OMI) on the Aura satellite are used to study the effect of energetic particle precipitation (EPP, as proxied by the geomagnetic activity index $A_p$) on the Antarctic stratospheric $NO_2$ column in late winter-spring (Aug-Dec) during the years 2005–2017. We show that the polar (60°S-90°S) stratospheric $NO_2$ column is significantly correlated with EPP throughout the Antarctic spring, until the breakdown of the polar vortex in November. The strongest correlation takes place during years with easterly phase of the quasi-biennial oscillation (QBO). The QBO modulation may be a combination of different effects: QBO is known to influence the amount of the primary $NO_x$ source, $N_2O$, via transport from the equator to the polar region. The QBO phase also affects polar temperatures, which may provide a link to the amount of denitrification occurring in the polar vortex. We find some support for the latter in analysis of temperature and $HNO_3$ observations from the Microwave Limb Sounder (MLS/Aura). Our results suggest that once the background effect of the QBO is accounted for, $NO_x$ produced by EPP significantly contributes to the stratospheric $NO_2$ column at the time and altitudes when the ozone hole is present in the Antarctic stratosphere. Based on our findings, and the known role of $NO_x$ as catalyst for ozone loss, we propose that as chlorine activation continues to decrease in the Antarctic stratosphere, the total EPP-$NO_x$ needs be accounted for in predictions of Antarctic ozone recovery.

## 1 Introduction

In the polar stratosphere, the dominant source of odd nitrogen, $NO_x$ (NO + $NO_2$), is produced via the oxidation of nitrous oxide, $N_2O$ (Brasseur and Solomon, 2005):

$$N_2O + O(^1D) \longrightarrow 2NO. \tag{1}$$

This reaction requires the presence of excited oxygen atoms $O(^1D)$, which are produced in the atmosphere by photolysis of ozone ($O_3$) and thus depend on sunlight being present. As a result, NO production via reaction (1) only takes place outside polar winter conditions. Following reaction (1) the existing NO can be converted to $NO_2$ in reaction with ozone:

$$NO + O_3 \longrightarrow NO_2 + O_2. \tag{2}$$

As N$_2$O production *in situ* in the polar stratosphere is insignificant, the polar stratospheric NO$_x$ production is highly dependant on the amount of N$_2$O transported from the tropics (Brasseur and Solomon, 2005). This principal source of polar N$_2$O is injected from the troposphere into the stratosphere at equatorial latitudes. It is then transported towards the polar regions by the large scale Brewer-Dobson circulation. Recent work by Strahan et al. (2015) has shown that the phase of the QBO influences the transport of N$_2$O from the surfzone to the polar vortex with a lag of 12 months. Further, their results (Figure 1 of Strahan et al. (2015)) indicate that easterly phase of the QBO during June-July is also generally associated with positive N$_2$O anomalies in the polar stratosphere between altitudes of ∼24-33 km in September, and opposite for westerly phase of the QBO. Notably for our study, these particular altitudes, at this time, are also affected by large scale transport of mesospheric air masses affected by energetic particle precipitation (Funke et al., 2014a).

The main pathway to NO$_x$ loss is via photolysis (Brasseur and Solomon, 2005). During polar night conditions when little to no sunlight is available, this results in long chemical lifetime (weeks to months) for the NO$_x$ family. However, NO$_x$ can be removed from the lower stratosphere during the polar night in a process known as denitrification which removes NO$_x$ when it is stored in the HNO$_3$ reservoir. This requires the winter vortex to be cold enough that polar stratospheric clouds (PSCs) form. Denitrification occurs when reactive nitrogen (particularly NO$_2$) is converted into HNO$_3$ in the lower stratosphere (Santee et al., 1995). HNO$_3$ is readily incorporated into PSCs, removing gaseous HNO$_3$ from the lower stratosphere as it eventually falls into the troposphere via gravitational sedimentation (Brasseur and Solomon, 2005).

## 1.1 EPP indirect effect

It is now well established that precipitating energetic particles can drive large enhancements in NO$_x$ quantities in the polar atmosphere (see e.g. Seppälä et al., 2007a; Funke et al., 2014a, b). Energetic particle precipitation (EPP) is the flux of charged particles (protons and electrons) of solar and magnetospheric origin into the Earth's atmosphere. The charged particles are guided to the polar regions by the Earth's magnetic field. Once reaching the atmosphere, they ionise the main neutral gases, N$_2$ and O$_2$. The chain of ion-neutral reactions that follows the ionisation then leads to increases in NO$_x$ species (this is known as "EPP-NO$_x$"), particularly in the mesosphere and lower thermosphere (Brasseur and Solomon, 2005). EPP manifests as energetic electron precipitation (EEP) as well as proton precipitation, which in the form of solar proton events (SPEs) is the most extreme from of EPP (see e.g. Seppälä et al., 2014). SPEs are usually associated with coronal mass ejections (CMEs) and thus, while the particles are highly energetic and have the ability to ionise as far down as the stratosphere, the events are short (hours to days) in duration, and occur sporadically. Conversely, EEP is always present in some form and is mostly dependant on solar wind speed (Funke et al., 2014b). Due to the lower energies of the electrons, EEP driven *in situ* NO$_x$ increases typically occur in the mesosphere and above (Turunen et al., 2009). When EPP occurs over the winter pole, the mesospheric NO$_x$ has a long chemical lifetime and can be transported downwards into the stratosphere inside the polar vortex. Once in the stratosphere, these NO$_x$ enhancements are effective at catalytically destroying ozone (see e.g., Jackman et al., 2008, and references therein). As NO$_x$ is not formed from N$_2$O during winter, EPP becomes a significant contributor to the polar winter NO$_x$ budget (Funke et al., 2014b).

## 1.2 Previous work

Several previous studies have examined the effects of EPP on polar winter $NO_x$, or the wider $NO_y$ family which includes both $NO_x$ and its reservoir species such as $NO_3$, $N_2O_5$, $HNO_3$, and $ClONO_2$. We will summarise the findings of the key observational works, with particular focus on those with SH or $NO_x$ transport aspects, in the following. Note that we will use the the the term $NO_x$ or $NO_y$ depending on which the study in question addressed.

Randall et al. (1998) reported stratospheric $NO_2$ observations from the Polar Ozone and Aerosol Measurement (POAM II) over three polar late-winter/early-spring periods. They found evidence to suggest that $NO_x$ from the SH polar mesosphere was transported down into the stratosphere inside the polar vortex during the winter. They also suggested that the observed enhanced levels of stratospheric $NO_x$ in 1994 could be, at least partially, due to production by EPP that took place at higher altitudes before the downwards transport. Based on their analysis, Randall et al. (1998) suggested that $NO_x$ transported to stratosphere from the mesosphere and above during the polar winter should be observable in Antarctic $NO_2$ column measurements.

Siskind et al. (2000) used Halogen Occultation Experiment (HALOE) observations from the UARS satellite between 1991-1996 to track $NO_x$ enhancements in October in the SH polar region. They found that the year-to-year variability in $NO_x$ inside the polar vortex followed variability in wintertime mean auroral $A_p$ index, a measure now frequently used for overall EPP levels (Matthes et al., 2017). At the time, they found that the peak $NO_x$ enhancements from "auroral" activity corresponded to around 3-5% of the total $NO_x$ generated from $N_2O$. Studies of $NO_x$ enhancements following the large Halloween SPEs in October-November 2003 also revealed large quantities of $NO_x$ in the Northern Hemisphere (NH) in January (Jackman et al., 2005). However, dynamical effects, driven by the following major sudden stratospheric warming (SSW), indicated that this $NO_x$ was unlikely to have originated from the SPEs (Seppälä et al., 2007b). The $NO_x$ increases were more likely a result of the the large amounts of EEP that was present during the polar winter combined with downward descent. For example, Randall et al. (2005) used observations from a number of satellite instruments to show that the springtime $NO_x$ increases in the NH were influenced by strong downward descent in January 2004, bringing excess amount of $NO_x$ down to the stratosphere. Randall et al. (2007) used Atmospheric Chemistry Experiment-Fourier Transform Spectrometer (ACE-FTS) together with HALOE observations to show that the peak upper stratospheric EPP-$NO_x$ was highly correlated with EEP levels over 1992–2005, up until September in the SH. Dynamics influencing the polar vortex is one of the main reason relations between $NO_x$ observations and EPP break down. This is particularly important in the NH, where the polar vortex is more susceptible to SSWs. Randall et al. (2007) further found that the largest EPP-$NO_x$ occurred during the declining phase of the solar cycle, during which more high-speed solar wind streams, driving EEP, are likely to occur.

Using tracer correlations to quantify EPP-$NO_x$ vs $NO_x$ from $N_2O$ oxidation, Randall et al. (2007) suggested that the maximum EPP-$NO_x$ enhancements made up to 40% of the total polar $NO_y$ (total reactive nitrogen, $NO_y$ = NO + $NO_2$ + $NO_3$ + $ClONO_2$ + $HNO_4$ + $N_2O_5$ + $HNO_3$) budget. Seppälä et al. (2007a) contrasted average wintertime $A_p$ levels with the polar winter upper-stratospheric-lower mesospheric $NO_x$ observations from Global Ozone Monitoring by Occultation of Stars (GOMOS) onboard the Envisat satellite, finding a nearly linear relationship between the two during 2002-2006 for both hemispheres. Funke et al. (2014b) used Michelson Interferometer for Passive Atmospheric Sounding (MIPAS) observations, also

from Envisat, of $NO_y$, to quantify the amount of EPP-$NO_y$ in the polar winter. Analogous to Randall et al. (2007), they found that EPP-$NO_y$ accounted for up to 40% of the wintertime polar $NO_y$. Funke et al. (2014a) then correlated $A_p$ and EPP-$NO_y$ in the wintertime and concluded that the strong relationship between $A_p$ and EPP-$NO_y$ supports using $A_p$ as a proxy for tracking EPP-$NO_y$ production in the SH wintertime.

## 1.3 This work

Here, we use stratospheric $NO_2$ column observations from the Ozone Monitoring Instrument (OMI) on-board the Aura satellite to investigate EPP as a source of $NO_x$ variability in the Antarctic later-winter - spring. We have a relatively long satellite period (2005-2017) in which to analyse how this $NO_x$ propagates in the following springtime, and whether this is detectable in the $NO_2$ column. We also analyse how the phase of the QBO affects the contribution of EPP-$NO_2$ to the total $NO_2$ column in springtime. The QBO influence could be likely due to a combination of two effects: 1) QBO phase influences transport of $N_2O$ to the polar region (Strahan et al., 2015) resulting in a increased background $NO_x$ source at key times and key altitudes during easterly QBO years (opposite for westerly QBO); 2) QBO conditions influence polar temperatures, which may affect the probability of PSC formation. Since PSC formation is linked to denitrification, there may be a connection between removal of $NO_x$ and QBO. To test for the latter, we will provide a simple analysis of $HNO_3$ and temperature observations from the Microwave Limb Sounder (MLS), also on-board Aura (more detailed analysis of PSC observations would be required to test this further).

## 2 Data sets

### 2.1 OMI $NO_2$ observations

We use stratospheric $NO_2$ column observations from the Dutch-Finnish built Ozone Monitoring Instrument (OMI) on-board NASA's Aura satellite from August to December during years 2005–2017 (v3, Level 2 daily gridded $NO_2$, see Krotkov (2012); Krotkov et al. (2017)). The daily gridded data has $0.25° \times 0.25°$ horizontal resolution. In our analysis, we use data from latitudes poleward of 50°S. Note that all latitudes henceforth are geographic latitudes. Aura is in a Sun-synchronous orbit in the "A-train" constellation (orbital altitude of 705 km, inclination of 98°, 16 day repeat cycle), with ascending node crossing the equator approximately at 1:45pm daily (Schoeberl et al., 2008). As a result, OMI measurements take place at the same locations each year. While $NO_2$ notably has a diurnal cycle, using observations from the same sunlit locations (thus same local times) each year minimises the effect of this in our analysis.

The OMI-$NO_2$ data is provided as total column, as well as separated tropospheric and stratospheric columns. This separation is based on the location of the tropopause. Here, we use the OMI stratospheric column observations only. The effective vertical range of the stratospheric column based on the OMI averaging kernels corresponds to $\sim 15 - 35$ km. At these altitudes, $NO_2$ makes up about 80% of the total $NO_x$ during daytime (see Brasseur and Solomon, 2005, chapter 5.5). As this corresponds to a large fraction of the total $NO_x$, we take the OMI $NO_2$ column measurements to represent a reasonable proxy for the variation

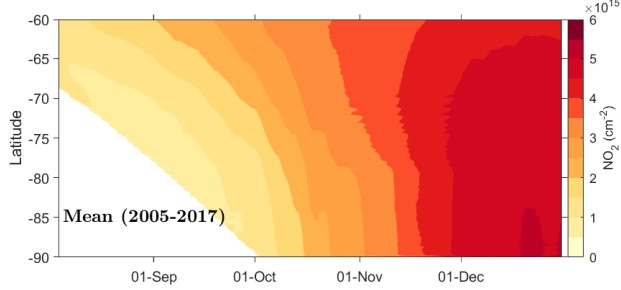

**Figure 1.** OMI 3 day running mean zonally averaged $NO_2$ column for the time period 2005–2017. The contour interval is $0.5 \times 10^{15}$ cm$^{-2}$. The white area at high latitudes in August-September indicates polar night conditions where OMI observations are not available.

in total $NO_x$. The algorithm for the OMI column separation is described in Bucsela et al. (2013). OMI measures back-scattered
solar radiation from the atmosphere. Thus, observations are only available for solar illuminated locations – there is no coverage
during polar night conditions. The latitudinal coverage is illustrated in Figure 1, which presents the zonally averaged mean $NO_2$
column for the period under investigation (2005–2017). The figure shows how the $NO_2$ column varies in the polar springtime,
with increasing amounts of $NO_2$ in the stratosphere as time progresses due to release from its reservoirs (Dirksen et al., 2011).
The error in the individual $NO_2$ column measurement is estimated to be $< 2 \times 10^{14}$ molecules cm$^{-2}$, however in areas with
low levels of tropospheric pollution (such as the Southern polar region), this error is considerably less (Bucsela et al., 2013).
Since June 2007, OMI-$NO_2$ has experienced an issue known as the row anomaly (RA) affecting certain fields of view. All RA
affected measurements have been excluded here, leaving around $2 \times 10^5$ observations poleward of $60°$S per day for the analysis
period.

    The Aug-Dec monthly mean polar ($60°$S to $90°$S) average zonal mean $NO_2$ columns for each year are listed in Table 1.

## 2.2    MLS observations

We use $HNO_3$ and temperature profiles from NASA's Microwave Limb Sounder (MLS) which is also on-board the Aura
satellite (Manney et al., 2015; Schwartz and Read, 2015). This study uses the version 4.2 product with data screened according
to Livesey et al. (2017). The latitude range used here is $60°$S to around $82°$S and the pressure range used is approximately
100 hPa to 10 hPa. MLS $HNO_3$ profiles have been validated by Santee et al. (2007), using data from both the $HNO_3$ 240–
135    GHz radiometer (for pressures $\geq 22$ hPa) and $HNO_3$ 190–GHz radiometer (for pressures $\leq 15$ hPa). MLS $HNO_3$ has vertical
resolution of 3-4 km in the lower – middle stratosphere (used here) and the precision of individual profiles is around 0.6 ppbv
in this region. The estimated error in these profiles is no more than 10%. MLS temperature profiles have been validated by
Schwartz et al. (2008) with precision of around $\pm 0.6$ K on individual profiles between 100 hPa and 10hPa.

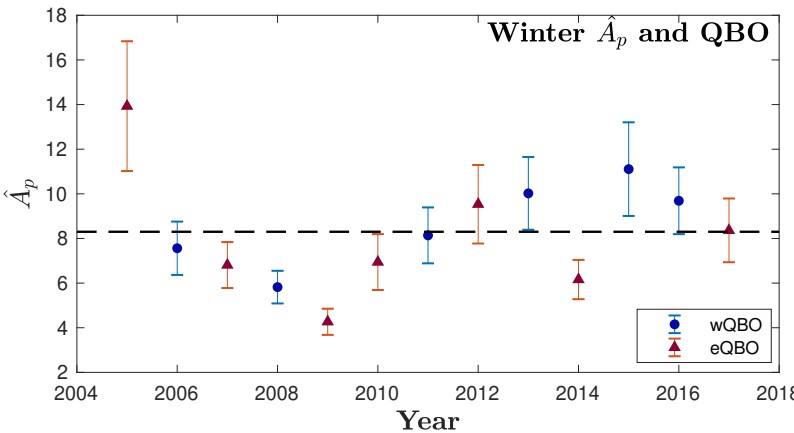

**Figure 2.** Mean wintertime $A_p$ ($\hat{A}_p$) for each year of this study with error bars indicating 2 times the standard error in the mean. Dashed line indicates the average $\hat{A}_p$ for this study (8.3). The direction of the QBO at 25 hPa in May is indicated with red triangles representing eQBO and blue circles wQBO.

## 2.3 EPP proxy

The geomagnetic activity index $A_p$ is a well-established proxy for EPP (see e.g. Matthes et al., 2017; Funke et al., 2014a) and is used here to estimate the overall levels of EPP for each polar winter under investigation. During 2005–2017, the mean of winter $A_p$ was 8.3, reflecting the relatively low overall solar activity during solar cycle 24 (solar cycle 23 average was 12.9). To estimate the overall EPP activity during each winter, we calculate the mean $A_p$ for the period of May–August of each year. These means are hereafter referred to as $\hat{A}_p$ and are provided in Table 1. We designate high $\hat{A}_p$ (H-$\hat{A}_p$) winters as those with $\hat{A}_p > 8.3$, i.e. $\hat{A}_p$ higher than the average for 2005–2017. Similarly, we take low $\hat{A}_p$ winters (L-$\hat{A}_p$) as those with $\hat{A}_p < 8.3$. The variation in winter $\hat{A}_p$ throughout this study is shown in Figure 2. This figure captures the 11-year solar cycle fairly well, with minimum around 2009 and maximum around 2015.

Several previous works have used tracer correlations to extract solely EPP produced $NO_x$ (or $NO_y$ when information on the $NO_x$ reservoirs is available) (see e.g. Randall et al., 2007; Funke et al., 2014b). When tracer information is not available, other works have investigated the variability in odd nitrogen resulting in variability in EPP levels (usually proxied by $A_p$) (see e.g. Seppälä et al., 2007a). Here, we focus on finding evidence of EPP contribution to column observations. As we do not have mesosphere-stratosphere descent tracer observations available from OMI, we are unable to use the tracer correlation methods. To overcome this we perform correlation analysis both for latitudinal coverage and polar average $NO_2$ observations to find evidence of $A_p$ driven variability in the Antarctic $NO_2$ column. All correlations between the $NO_2$ columns and $\hat{A}_p$ are based on the Spearman rank correlation (Spearman $\rho$), as it more robustly accounts for any non-linear relationships (Wilks, 2011) while still interpreting linear trends where present. Statistical significance is here defined as correlations significant at $\geq 95\%$ (i.e. $p$-value of $\leq 0.05$).

**Table 1.** May-Aug mean $A_p$ ($\hat{A}_p$) $\pm 2\times$ standard error in the mean, the polar (60°S to 90°S) Aug-Dec monthly mean, $\cos(latitude)$ weighted, zonal mean stratospheric $NO_2$ column density ($\times 10^{15}$ cm$^{-2}$), and QBO phase (E for easterly, W for westerly) for each year 2005-2017.

| Year | $\hat{A}_p$ | Aug $NO_2$ | Sep $NO_2$ | Oct $NO_2$ | Nov $NO_2$ | Dec $NO_2$ | QBO |
|------|-------------|------------|------------|------------|------------|------------|-----|
| 2005 | 13.9 ($\pm$2.9) | 0.43 | 0.49 | 0.84 | 1.16 | 1.24 | E |
| 2006 | 7.6 ($\pm$1.2) | 0.59 | 0.37 | 0.69 | 0.96 | 1.22 | W |
| 2007 | 6.8 ($\pm$1.0) | 0.41 | 0.47 | 0.82 | 0.98 | 1.18 | E |
| 2008 | 5.8 ($\pm$0.7) | 0.35 | 0.37 | 0.77 | 0.99 | 1.20 | W |
| 2009 | 4.3 ($\pm$0.6) | 0.40 | 0.42 | 0.77 | 1.00 | 1.25 | E |
| 2010 | 6.9 ($\pm$1.3) | 0.43 | 0.45 | 0.78 | 1.00 | 1.14 | E |
| 2011 | 8.1 ($\pm$1.3) | 0.37 | 0.39 | 0.66 | 0.96 | 1.18 | W |
| 2012 | 9.5 ($\pm$1.8) | 0.46 | 0.51 | 0.90 | 1.09 | 1.19 | E |
| 2013 | 10.0 ($\pm$1.6) | 0.39 | 0.46 | 0.87 | 1.15 | 1.23 | W |
| 2014 | 6.2 ($\pm$0.9) | 0.39 | 0.46 | 0.76 | 1.03 | 1.22 | E |
| 2015 | 11.1 ($\pm$2.1) | 0.37 | 0.40 | 0.63 | 0.98 | 1.13 | W |
| 2016 | 9.7 ($\pm$1.5) | 0.41 | 0.44 | 0.75 | 1.07 | 1.16 | W |
| 2017 | 8.4 ($\pm$1.4) | 0.41 | 0.46 | 0.74 | 1.09 | 1.19 | E |

## 2.4 Quasi Biennial Oscillation

To investigate the potential QBO effect in the Antarctic atmosphere, we estimate the phase of the QBO from the 25 hPa level
zonal mean zonal wind (Naujokat, 1986) near the equator in May each year. For use of the 25 hPa level in the Southern
Hemisphere, see Baldwin and Dunkerton (1998). We designate years when the zonal mean zonal wind direction is easterly as
easterly QBO (eQBO), and westerly as westerly QBO (wQBO). The QBO direction for each year of the study is indicated in
both Table 1 and Figure 2. Figure 2 illustrates the approximately biennial nature of the oscillation, with the direction changing
almost every year.

## 3 Results

### 3.1 $NO_2$ anomalies

We first investigate the anomaly from the mean for each of the four different categories of this study, eQBO H-$\hat{A}_p$, eQBO
L-$\hat{A}_p$, wQBO H-$\hat{A}_p$, and wQBO L-$\hat{A}_p$. This is to show how $NO_2$ column evolves in the springtime in the different conditions
and to further justify the splitting of years based on QBO phase. Figure 3 presents the average anomaly, (i.e. the mean as in
Figure 1 deducted) for each of the four different categories of this study. We can see that winter $\hat{A}_p$ affects the column $NO_2$
present in the spring: years with H-$\hat{A}_p$ (panels a and c) having more positive anomalies from August to November especially in

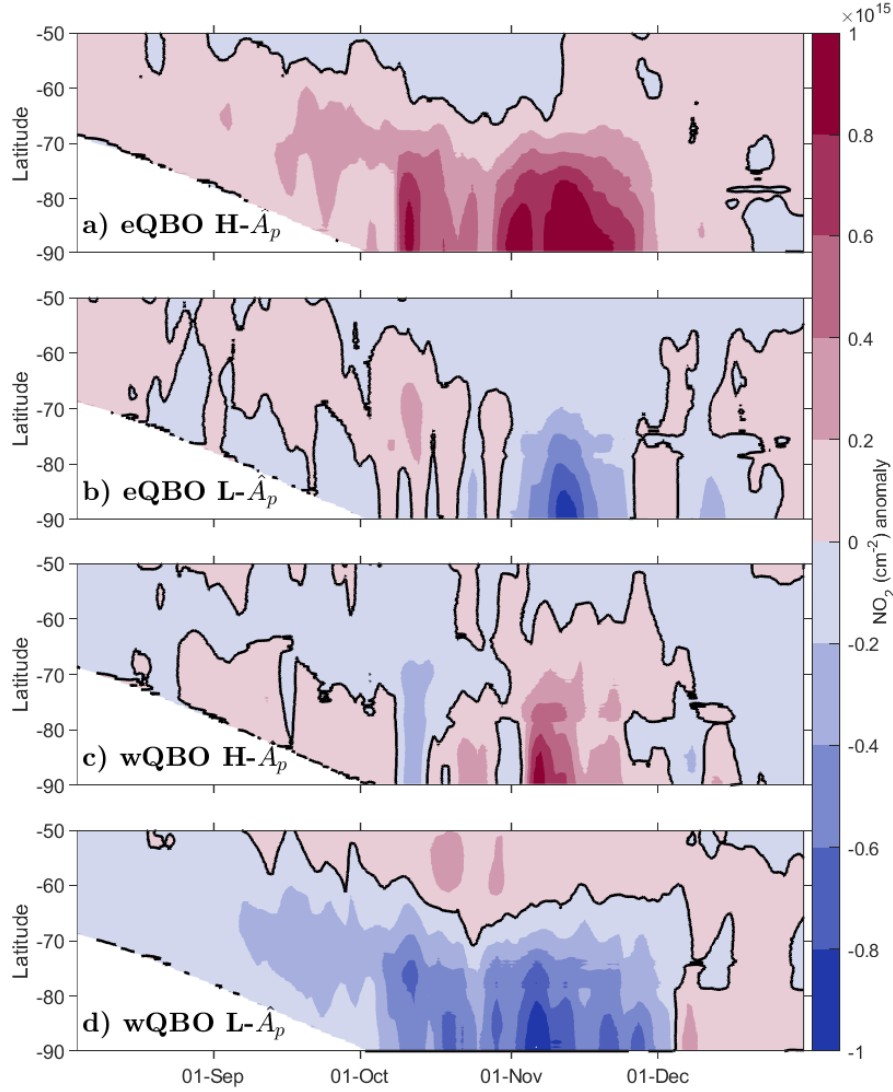

**Figure 3.** a) mean $NO_2$ anomaly for years with both H-$\hat{A}_p$ and eQBO. Contour level is $0.2\times10^{15}$ cm$^{-2}$ with black contour representing zero anomaly. b)-d) as b) but different combinations of $\hat{A}_p$ and QBO (see Figure).

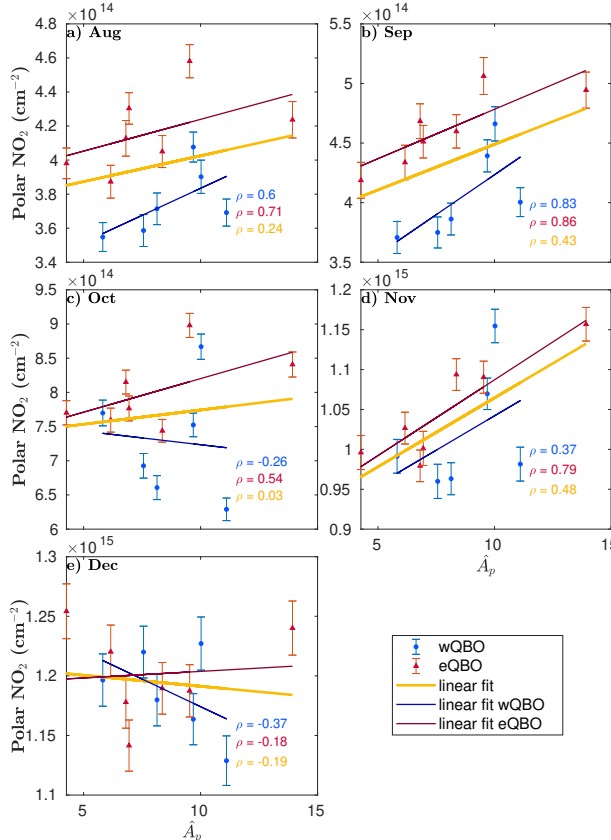

**Figure 4.** $\hat{A}_p$ versus monthly average, $\cos(latitude)$ area weighed $NO_2$ column density averaged over $60°$S to $90°$S (where available, see text) for the months of August to December (months as shown in each individual panel). Red triangles indicate years with eQBO and blue circles years with wQBO. The yellow line shows a least squares linear fit to all data, red line eQBO years only, and blue line wQBO years only. The Spearman $\rho$ (correlation coefficient) for each set is shown in each with corresponding colour, e.g. red $\rho$ corresponds to correlation coefficient for eQBO years etc. Error bars are $2\times$ the standard error in the mean.

the highest latitudes. In b) and c), the month of October is highly variable with both showing regions of positive and negative anomaly. For low $\hat{A}_p$ years (panels b and d) early spring is not consistently positive or negative, however November displays negative anomalies at high latitudes. The combined influence of QBO and $\hat{A}_p$ appears to be most significant for H-$\hat{A}_p$ eQBO

175   years, and wQBO L-$\hat{A}_p$ years (panels a and d). These show consistent but opposite behaviour throughout the spring, with H-$\hat{A}_p$ eQBO years the most favourable for $NO_2$ and wQBO L-$\hat{A}_p$ the least.

## 3.2   Mean SH polar columns

Figure 4 presents the $\hat{A}_p$ and the mean polar ($60°$S–$90°$S) $NO_2$ column for each year (2005-2017) and for each individual month from August to December. This is to investigate whether there is a relationship $\hat{A}_p$ size and $NO_2$ column, and how

this is affected by QBO phase. The mean polar columns are area weighted by cosine of latitude. The phase of the QBO in the preceding May is indicated with red triangles corresponding to eQBO and blue circles to wQBO conditions. Least squares linear fits for all years, wQBO, and eQBO years are included in each panel to guide the eye. The Spearman correlation coefficient ($\rho$) for each month for all years (yellow) along with eQBO (red) and wQBO (blue) years only are also included in the panels. Note that, as the OMI measurement field gradually increases from an initial maximum latitude of around 68°S in August to 90°S by the end of September, the total $NO_2$ column values do not initially fully encompass the entire polar region (60°S–90°S). The missing data in August and September is treated as missing values in the mean calculation and as such does not contribute to the mean in these figures.

The results shown in Figure 4 suggests that a correlation between $\hat{A}_p$ and the stratospheric $NO_2$ column occurs in August, September and November. This is consistent with Figure 3. Furthermore, there is a clear positive correlation for eQBO years from August to November, while for wQBO years, the positive correlation in August and September disappears in October. While the wQBO November linear fit is close to the total fit, the individual years show large variability. In general, wQBO years have consistently lower column $NO_2$ values, especially in August and September. The generally reduced levels of $NO_2$ during wQBO conditions is compatible with the analysis of Strahan et al. (2015), which indicated that the altitudes where Funke et al. (2014a) reported EPP-$NO_y$ enhancements in later winter-early spring, have consistently lower/higher levels of the dominant $NO_x$ source, $N_2O$, during wQBO/eQBO.

To contrast our results with previous extensive work by Funke et al. (2014a) we repeated the analysis presented in Figure 4 using the units of gigamole (Gmol, see Funke et al. (2016)) for the monthly mean polar $NO_2$ columns). This figure is included in the appendix as Figure A1 and shows the least squares linear fits, with the corresponding parameters given in each panel. This allows us to estimate the EPP contribution to the lower stratosphere ($\sim 15 - 35$ km) $NO_2$ in the spring, analogous to Funke et al. (2014a). For example, in September (Figure A1, panel b), the approximate contribution to polar stratospheric $NO_2$ column from EPP in eQBO years is $+0.023$ Gmol/$\hat{A}_p$. The largest contribution to OMI lower stratospheric $NO_2$ from EPP occurs in November, with the corresponding values of $+0.073$ (eQBO), $+0.072$ (wQBO) and $+0.066$ (all years) Gmol/$\hat{A}_p$, respectively. Funke et al. (2014a) (see their Figure 10, showing excess EPP-$NO_y$ in the stratosphere – lower mesosphere in early spring) found an increase in SH polar EPP-$NO_y$ of around $+0.0698$ Gmol/$A_p$ unit in September. Contrasting this to our results of $+0.066$ Gmol/$\hat{A}_p$ in November, it seems a large fraction of the EPP-$NO_y$ detected by Funke et al. (2014a) is maintained in the polar region and able to reach the lower stratosphere where it can be detected as $NO_2$ still in November. Note that Funke et al. (2014a) use a weighted $A_p$ scheme.

### 3.3 Latitudinal correlations

Figure 5 shows the latitudinal extent of the correlation between $\hat{A}_p$ and the 7 day running mean $NO_2$ column for latitudes 50°S–90°S averaged over 1° latitude bins. Stippling indicates correlation significant at $\geq 95\%$ level. This shows how correlation between $\hat{A}_p$ and $NO_2$ column evolves over time and latitude, and different QBO phase. Panel a) presents the correlation when all years are taken into account. Significant positive correlation occurs in late August and variably throughout September, then again in November. October and December show little to no significant correlation. Panel b) shows the correlation for

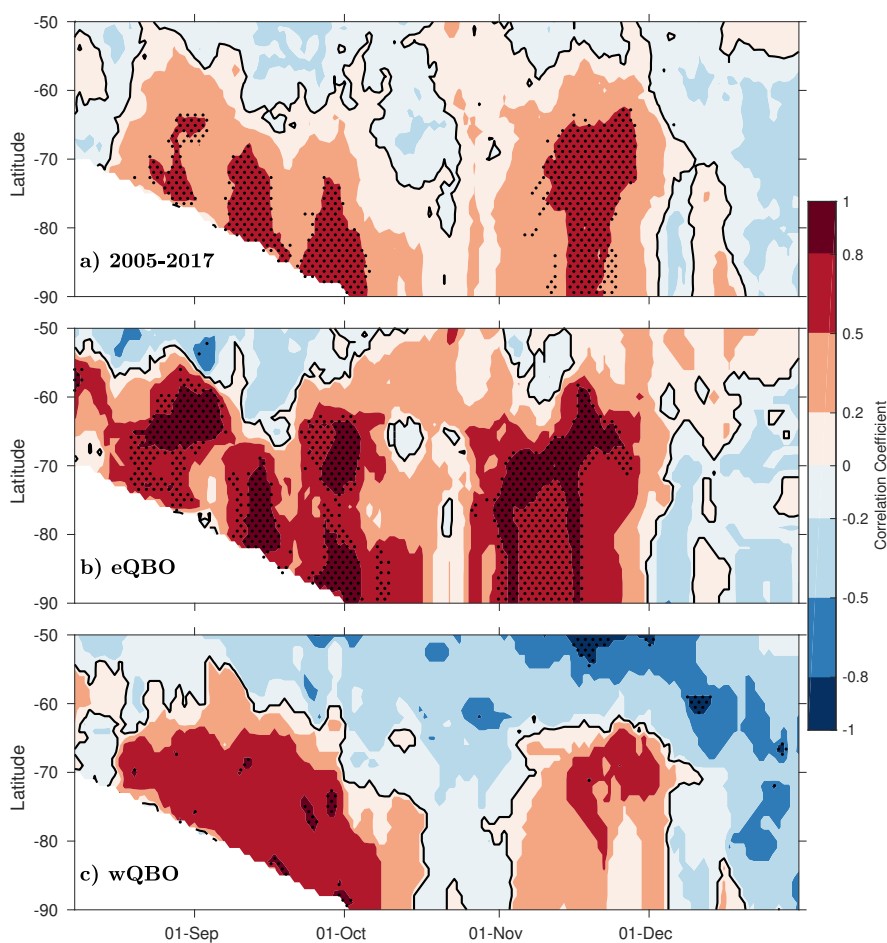

**Figure 5.** a) Correlation of $\hat{A}_p$ and 7 day running mean NO$_2$ column density for Aug-Dec for all years. b) Correlation of $\hat{A}_p$ and 7 day running mean NO$_2$ column density for years with eQBO only. c) as b) except for years with wQBO only. All figures have $1^o$ latitude resolution. Contour levels are shown for [-1, -0.8, -0.5, -0.2, 0, 0.2, 0.5, 0.8, 1] and the zero contour is indicated with black. Stippling shows regions of correlation significant at $\geq$95% level.

eQBO years only. There are areas of statistically significant positive correlations in all months except December. In October, significant correlations occur only at the very beginning of the month. High positive correlations are still present across 60°S to 90°S from early to mid-November providing first evidence of the indirect EPP-NO$_x$ effect lasting well into the SH spring season. Figure 5 c) presents correlation for years with wQBO only. While positive correlations are present throughout August and September, only small regions are found to be statistically significant. October marks a shift towards negative correlation (not statistically significant) at all latitudes. In November the correlations turn positive once more, but these are again, not statistically significant.

The results shown in Figure 5 suggest that the NO$_2$ increases at high polar latitudes in September are due to increased EPP/geomagnetic activity, as strong correlations between NO$_2$ and $\hat{A}_p$ occur in all panels. They also imply that increases in NO$_2$ in November can be due to a combination of high EPP activity and eQBO, whereas wQBO appears to reduce the occurence of any EPP induced NO$_2$ increases.

## 3.4 Polar vortex influence in October

The correlations presented in the previous sections were found to have fewer occurrences of statistical significance in October than the surrounding months. As this time of year marks the typical breakup period of the polar vortex (Hurwitz et al., 2010) and knowing that the descent of EPP produced NO$_x$ is limited to inside the polar vortex (as previously demonstrated for October by Siskind et al. (2000)), we will now investigate October separately, taking the polar vortex into account.

To account for effects from potential asymmetries in the shape of the polar vortex in October in our zonal mean calculations, we repeated the earlier analysis for measurements located inside the vortex. To establish the location of the edge of the polar vortex we utilised the OMI co-located ozone column measurements (Bhartia, 2012): Ozone depleted air is isolated within the vortex until the vortex break up, typically in late November (Kuttippurath and Nair, 2017). Based on this, we take measurement locations poleward of 50°S with corresponding stratospheric ozone column of $< 245$ DU to be inside the polar vortex. We perform this method for every day of October in the study period to find the daily vortex extent. This is then used to locate NO$_2$ that is inside the vortex for every day in October over the study period. An example of how the ozone column and the estimated vortex edge are reflected on the NO$_2$ column measurement for one day of the study (19th October 2014) is shown in Figure 6. Note that we are unable to use this method on months prior to October due to OMI's viewing method and the ozone column not being sufficiently low to detect a clear vortex edge.

Figure 7 a shows the October results (as in Figure 4 c) when only observations inside the polar vortex are included. We find that for eQBO, the observations are now much closer to the linear fit than in Figure 4 c), implying that the earlier disappearance of correlation was likely due to variations in the shape of the polar vortex in October.

Similarly for the horizontal distribution of the correlations (Figure 7 b), we now find high correlations for eQBO years throughout October. This again implies that the lack of correlation in October is due to the distorted shape of the polar vortex being smeared out by calculation of zonal means, and the effect of EPP on the NO$_2$ column is significant through October. The reappearance of correlations in eQBO years in November in Figure 5b) is likely a mixing effect, with the break down of the polar vortex around this time leading to vortex air being mixed with extra-vortex air resulting in the NO$_2$ distribution

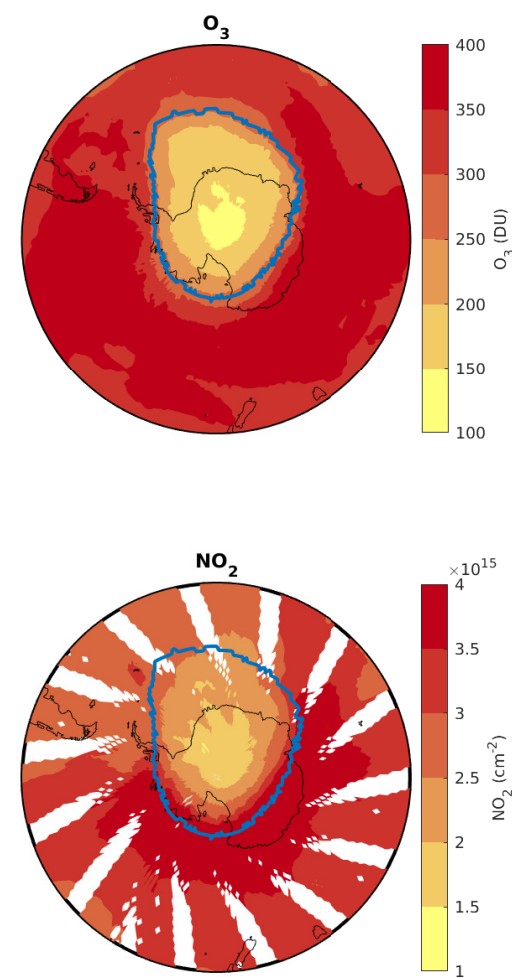

**Figure 6.** Vortex edge identification based on OMI ozone column for the 19th October 2014. The top panel shows the OMI measured ozone column density with the 245 DU contour highlighted. The bottom panel shows the $NO_2$ stratospheric column with the 245 DU ozone contour overlaid. This method is repeated for each individual day in October throughout the study period.

not being as skewed as it was when contained in the vortex. Figure 7 c also shows higher correlation with more instances of significance in wQBO years in October than in Figure 5 c though this is more variable than in eQBO years (which is consistent with Figure 5, that wQBO years show lower correlation). Although the $A_p$ index has generally been lower in the past decade than in the 1991-1996 period investigated by Siskind et al. (2000), considering observations only in the vortex still shows the same, strong linear relationship found in that study.

## 4   Discussion

### 4.1   Influence of the QBO

As shown in Figures 3, 4 and 5 a), our results suggest that the phase of the QBO is influencing the SH polar EPP-NO$_x$ signal in the spring months. Results of Strahan et al. (2015) suggest that eQBO phase in early winter leads to increased N$_2$O between altitudes of ~24-33 km in September: Figure 1 of Strahan et al. (2015) shows a positive anomaly for N$_2$O in September in eQBO years (and opposite for wQBO). Although their designation of QBO phase differs slightly from ours, this will not largely affect their comparability, as our designations only differ for one year of a 10 year study. Our results suggest that although there is an increased pool of N$_2$O in the polar region during eQBO (contributing to larger NO$_2$ column from N$_2$O oxidation), EPP still contributes a clear fraction to the overall SH polar stratospheric column NO$_2$ in the springtime. Taking the QBO phase into account this EPP contribution during spring is generally more pronounced. We note that for wQBO years, the high correlation with $\hat{A}_p$ is only present until September, while lasting until November for eQBO years. It is not clear why this is the case for wQBO. Further analysis of possible reasons behind this discrepancy is beyond the scope of this work.

### 4.2   Possible influence of PSCs and denitrification

Here we discuss possible reasons for the consistently lower amounts of NO$_2$ in wQBO years found in Figure 4. Baldwin and Dunkerton (1998) found that the polar vortex is colder during winters with wQBO. A colder polar vortex results in a higher likelihood of polar stratospheric cloud (PSC) formation (Brasseur and Solomon, 2005). As discussed in the Introduction, PSCs affect the heterogeneous chemistry in the polar region, leading to denitrification of the lower stratosphere (Dirksen et al., 2011). One of the possible reasons for the lower NO$_2$ column during wQBO found here could be enhanced denitrification as a result of lower temperatures in the vortex resulting in enhanced PSC formation. In the following we will perform a simple analysis to test for signs of this from existing MLS observations, although more detailed analysis of PSC observations would be required for robust conclusions.

To test whether QBO phase affects denitrification in the Antarctic stratosphere, we analysed temperature and HNO$_3$ observations from MLS (see section 2.2). Figure 8 a) and b) show the mean temperature and HNO$_3$ respectively, each averaged over 60°S to 82°S for 2005-2017 over the late winter–early spring period, i.e. when the polar vortex is coldest and PSCs are forming. Panels c) and e) show the anomalies from the mean temperature, for eQBO and wQBO years respectively. Panels d) and f) present the anomaly from the mean HNO$_3$ mixing ratio, for eQBO and wQBO years respectively. The vertical pressure

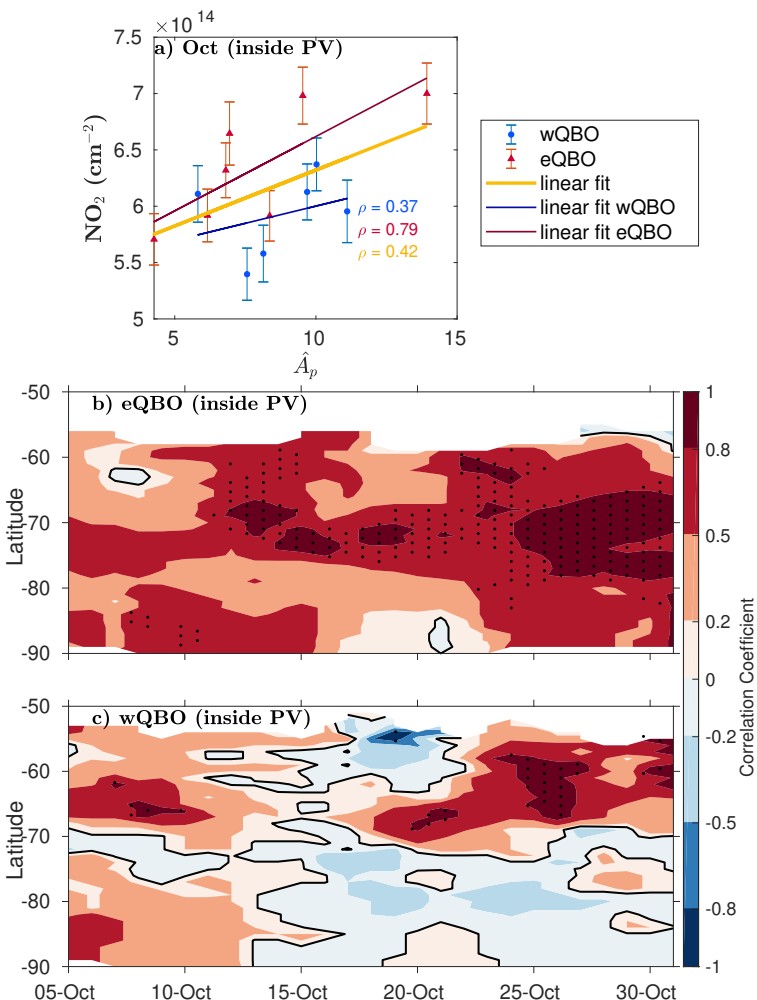

**Figure 7.** a) $\hat{A}_p$ vs. the average, $\cos(latitude)$ area weighted $NO_2$ from $60°$S to $90°$S, for observations inside the polar vortex. Red triangles correspond to eQBO years and blue circles to wQBO years. The yellow line represents a linear fit to all data points, while the red line is for eQBO years only and blue for wQBO years only. $\rho$ values are as in Figure 4. Error bars indicate the 95% confidence interval for the mean. b) Correlation of $\hat{A}_p$ with 5 day running mean $NO_2$ column inside the polar vortex for eQBO years. Contour levels are shown for [-1, -0.8, -0.5, -0.2, 0, 0.2, 0.5, 0.8, 1], with an additional black line for the zero contour. The stippling indicates correlations significant at $\geq 95\%$ level. c) as b) but for wQBO years.

range of all panels is 100 hPa to 10 hPa which corresponds to an altitude range of approximately 17 km to 32 km. Figure 8
suggest that eQBO years tend to have more $HNO_3$ (up to 1 ppbv) and higher temperature (up to 4 K) throughout this period, while wQBO years show a consistently negative anomaly in $HNO_3$ (down to $-1$ ppbv) and lower temperature (down to $-4$ K). Colder temperatures would likely lead to more PSC formation and thus more $HNO_3$ being removed from the stratosphere

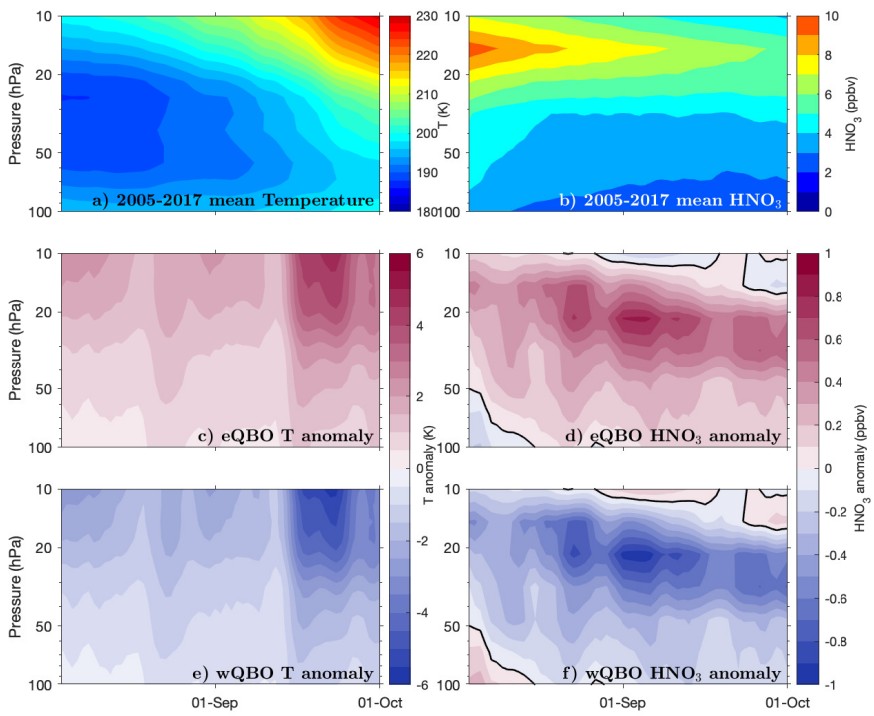

**Figure 8.** a) Three day running mean MLS temperature from 2005-2017 (5 K contour interval), b) three day running mean MLS HNO$_3$ from 2005-2017 with contour interval 1 ppbv. a) and b) are both averaged over 60°S to 82°S for the lower stratosphere for 2005-2017 late winter–early spring. c) HNO$_3$ anomaly (mean shown in a) has been subtracted) for years with eQBO (contour interval is 0.1 ppbv, with black contour showing zero anomaly). d) Temperature anomaly (mean shown in b) has been subtracted) for years with eQBO (contour interval is 1 K, with black contour showing zero anomaly). e) as c) but years with wQBO, and f) as d) for years with wQBO.

(more denitrification) in wQBO years (than in eQBO years). It should, however, be noted here that the link of PSC coverage and QBO modulation of polar temperature via the Holton-Tan effects is still under debate (see e.g. Strahan et al., 2015).

**5 Conclusions**

We have, for the first time, provided evidence of EPP contribution to the Antarctic stratospheric NO$_x$ column in the late springtime using OMI/Aura stratospheric NO$_2$ observations. This is one of the few studies to use stratospheric NO$_2$ data from OMI and also highlights the value of long time series of stratospheric NO$_2$ from nadir viewing instruments. Our analysis shows that influence from the QBO is able to mask the stratospheric EPP-NO$_x$ signal in satellite observations in a way that to our 290 knowledge has not previously been accounted for: Accounting for the phase of the QBO makes the contribution from EPP

more pronounced in the $NO_2$ column, and signals of enhanced EPP-$NO_x$ in the polar stratospheric column can be detected until late November.

Previously, Funke et al. (2014a) analysed EPP-$NO_y$ observations and were able to attribute SH enhancements with an average $A_p$ dependence of $+0.0698$ Gmol/$A_p$ into early spring months (September). Here, we show that this reactive nitrogen lingers, entering the lower stratosphere in the form of $NO_2$ at an average $A_p$ dependence of $+0.066$ Gmol/$\hat{A}_p$ in November.

We present evidence of contribution from EPP-$NO_x$ in the Antarctic stratosphere at a time when halogen activate ozone loss is taking place. $NO_x$ is well known to react with both ozone and active halogens, catalytically destroying the former, and driving the latter to its reservoirs (Brasseur and Solomon, 2005). Antarctic ozone loss has been found to be reduced in years with eQBO (Garcia and Solomon, 1987) due (at least in part) to the increased vortex temperatures hampering chlorine activation on PSCs (Lait et al., 1989). Our results suggest that, as chlorine activation continues to decrease in the Antarctic stratosphere following the Montreal Protocol (Solomon et al., 2016), the total EPP-$NO_x$ (in addition to SPEs as pointed out by Stone et al., 2018) should be accounted for in predictions of Antarctic springtime ozone recovery. Future studies should investigate the role of larger EPP-$NO_x$ fraction when investigating the net effect of $NO_2$ on the fragile ozone chemistry in the springtime.

*Data availability.* All data used here are open access and available from the following sources: $A_p$: http://wdc.kugi.kyoto-u.ac.jp/kp; QBO: https://www.geo.fu-berlin.de/en/met/ag/strat/produkte/qbo; OMI and MLS: https://earthdata.nasa.gov.

*Author contributions.* EMG and AS planned the study. EMG did the analysis with support from AS. JT provided expertise on OMI observations. EMG and AS lead the writing of the manuscript with comments from all authors.

*Competing interests.* The authors declare no competing interests.

*Acknowledgements.* We are grateful for the World Data Center for Geomagnetism, the Freie Universität Berlin and the National Aeronautics and Space Administration for providing open access to the data sets used in this study. We would like to thank Dan Smale and Cora Randall for their useful discussions and valuable feedback.

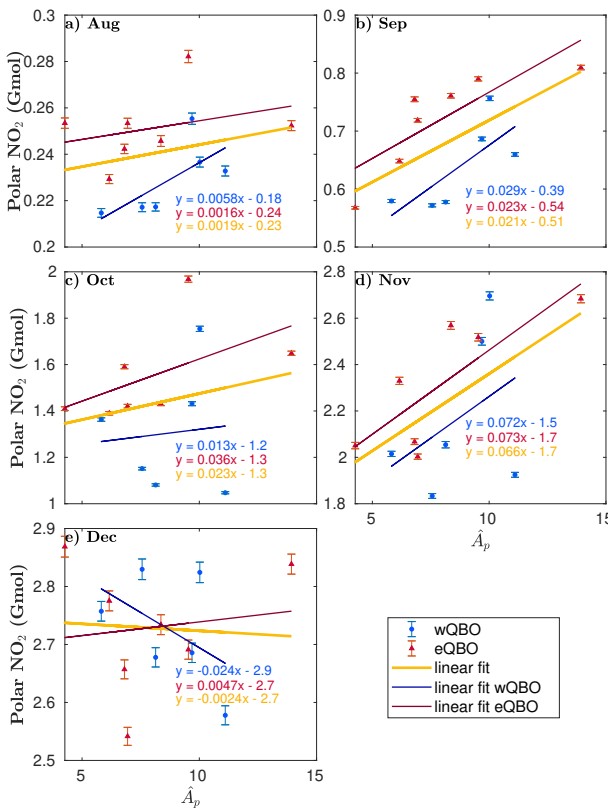

**Figure A1.** As Figure 4, but with monthly average polar NO$_2$ expressed in gigamole (Gmol). Each panel shows the linear least squares fit to data points (colour coding as before), including the fit equations ($y =$NO$_2$, $x = \hat{A}_p$). Red triangles are years with eQBO, and blue circles are wQBO. The yellow linear fit is a best-fit line for all the data in each plot, while the red fits only eQBO data, and the blue line fits only wQBO data.

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
