# Peer review of "Evidence for EPP and QBO modulations of the Antarctic NO2 springtime stratospheric column from OMI observations"

_Atmospheric Chemistry and Physics, 2019_

## Referee Comment (RC1) · Anonymous Referee #1 · 8 Jan 2020

This paper uses 13 years of Aura-OMI data to study the influence of energetic particle precipitation (EPP) and the QBO on the Antarctic stratospheric springtime NO2 column. The authors show that the NO2 column is positively correlated with the geomagnetic Ap index (used as proxy for EPP) until November and that the strongest correlations take place during years with easterly phase of the QBO. This is an interesting paper which should be suitable for publication after addressing my comments below, mostly dealing with the explanations provided for the encountered QBO dependence.

General comments:

[Figure]

(1) The title of the paper suggests that polar springtime EPP-NOx is influenced by the QBO, however, none of the suggested mechanisms results in a modulation of the EPP contribution. Specifically, the authors suggest that (i) the "amount of the primary NOx source, N2O, transported into the polar regions" is affected by the QBO, and (ii) the "QBO affects the temperature of the polar vortex and thus the amount of denitrification". (i) would affect only the background NOx concentration (produced by N2O oxidation) and not the EPP contribution. (ii) would represent a total NOy loss mechanism (independently whether produced by EPP or N2O) and hence would not alter the relative EPP-NOx contribution. In the sense a title like "Evidence for EPP and QBO modulations of the Antarctic NO2 springtime stratospheric column from OMI observations" would be more appropriate.

(2) It is suggested that, during eQBO, there is a lack of N2O transported to the polar regions which, in turn, results in a more prominent EPP-NOx contribution and hence better correlation of the observed NO2 column with Ap. This hypothesis is based on Fig 1 of Strahan et al. (2015) indicating a polar springtime N2O depletion during eQBO around 400-600 K (corresponding to approximately 15-25 km) from MLS observations. However, NOy production by N2O oxidation occurs predominantly at higher altitudes (peaking around 30 km which corresponds to a potential temperature level of around 800K) where the MLS observation analysed by Strahan et al. show a N2O increase during eQBO from the equator to around 70S. It is thus more likely that the background NO2 column is enhanced rather than decreased during eQBO because of increased N2O oxidation in the subpolar regions. Note that this is also in consonance with the results shown in Figures 3 and 4.

(3) It is further suggested that the "QBO affects the temperature of the polar vortex and thus the amount of denitrification", resulting in smaller NO2 losses and hence increased NO2 during eQBO. The authors base this explanation on MLS HNO3 observations, indicating an HNO3 increase during eQBO in the 100-10 hPa range. However, it is not clear whether this increase is caused by reduced HNO3 losses (due to a warmer

vortex and hence reduced PSC formation) or due to increased productions (e.g. by increased $N_2O$ oxidation as mentioned above). In order to proof their "denitrification" hypothesis, the authors should demonstrate that the $HNO_3$ enhancements during eQBO are linked to temperature increases and/or PSC occurrence. In this context it is worth to mention that the link of PSC coverage and QBO modulation of polar temperature via the Holton-Tan effects is still under debate (see, e.g, Section 4 of Strahan et al., 2015).

Specific comments:

l23-25: Strahan et al. have shown that the lower stratospheric $N_2O$ anomaly at 450 K in the Antarctic polar springtime vortex correlates with the surfzone anomaly at 650 K 12 months earlier, the latter being characterized by enhanced $N_2O$ during eQBO.

l27: strictly speaking it is $HNO_3$ (not $NO_x$) being removed by denitrification.

l63: the major SSW occurred in January 2004 (not December 2003).

l85-86: This sentence is a repetition of what is stated in the preceding paragraph.

l87 "...whether this IS detectable...."

l147: It is the combined EPP and QBO influence which leads to the most prominent differences between H-Ap/eQBO and L-Ap/wQBO years.

l186: Figure 5 shows correlations, not $NO_2$ column increases.

l204-205: What about wQBO? Fig 7a suggests that correlations improve also for wQBO when considering vortex-only observations.

l206: Consider to add "(see Fig. 7b)"

l215-220: see general comment (2)

l223-236: see general comment (3)

l229: QBO direction -> QBO phase

l251: "average rate" implies a time dependence. "average Ap dependence" would be clearer.

l257: Why should total EPP-NOx only be accounted for in eQBO years?
* * *

---

## Referee Comment (RC2) · Anonymous Referee #2 · 16 Jan 2020

Gordon et al. 2019 present a study of the Antarctic springtime stratospheric NO$_2$ column from Aura/OMI measurements, and they correlate those columns to the geomagnetic Ap index and the QBO. According to the authors this is the first study to use the OMI NO$_2$ column data product in a study like this. As middle atmospheric physics and chemistry is part of ACP, the manuscript fits the scope of this journal. However, in my opinion some revisions are necessary before the manuscript fits the quality standards of ACP.

[Figure]

**General comments**

Although the data and methods used are sound and the authors have taken great care to be in line with earlier studies that have investigated the EPP indirect effect, some questions on the methods need to be clarified.

**1** Parts of the paper seem to be disconnected and there does not seem to be a clear logical thread to guide the reader. Some of the sections seem to focus on describing figures without explaining why they are relevant to the study and how they relate to the other sections. In particular Sect. 3.1 (esp. Fig. 3) seems unrelated and not relevant to the rest of the paper.

**2** There seems to be a mixture of terms throughout the manuscript, "$NO_x$", "$NO_y$", and "$NO_2$" seem to be used interchangeably. For example the title states "EPP-$NO_x$" whereas the study focuses on $NO_2$ only. Although according to Brasseur and Solomon, 2005, $NO_2$ makes up around 80% of $NO_x$ in the stratosphere, this fact should be noted. The EPP part is not defined at all, Funke et al. 2014a,b use tracer correlations and Randall et al. 2007 use $CH_4$–$NO_2$ correlations to identify *EPP*-$NO_y$, how do the authors discriminate between EPP and non-EPP $NO_2$? A clear definition of these terms and how the authors use them should be given in Sect. 2.

**3** Do the authors average the 3h Ap or the daily mean Ap? Although other studies use average Ap as well (e.g. Funke et al. 2014a), Ap does not follow a normal distribution and the authors should be aware of that when using the mean as an estimator. This non-normality manifests itself in a very skewed distribution and a large standard deviation, particularly the 3h values. Has this been considered in the correlation analysis? I suggest that the authors check that the mean is a valid estimator for the distribution or cite a relevant publication. I also suggest to present the $\hat{A}_p$ values with error bars in Figs. 2, 4, 7, and A1 and Table 1 (probably based on appropriate quantiles).

**4** Is the 60°–90° average area weighted? If it is, it should be stated somewhere, Sect. 2

seems the obvious place (l.112). If not, higher latitudes may be artificially amplified in the polar cap average column. And in that case a discussion would be needed to assess the possible differences when taking area weighting into account.

**5** In Sect. 4.1 the authors discuss the possible impact of out-of-vortex air on reducing the correlation between $\hat{A}_p$ and the $NO_2$ column in October. Why is this presented in the "Discussion" section and not the "Results" section? As the authors seem to have an indication about the actual vortex available to them, why isn't the whole study based on vortex averages instead of whole polar cap averages? That would remove the ambiguity of including non-EPP-$NO_2$ from horizontal transport/mixing in the polar cap average.

**Specific comments**

Abstract

- ll.9–11: Does it really contribute to $NO_2$ or is it just the fraction that changes due to a varying background? I suggest to rephrase these sentences to be clearer, for example how is it linked to the ozone hole? What is cause and what is the effect? See also my other comments below.

Introduction (Sect. 1)

- I believe the introduction would profit from some additional subsections, e.g.:

  • ll.33–48 "EPP indirect effect"

  • ll.49–83 "Previous work"/"Earlier studies"

- ll.84ff "This work"

- ll.85–86: This is a repetition and can be removed.

- l.87: A verb is missing: "... this *is* detectable ..."

- The authors do not mention in the introduction that they are going to use (MLS) $HNO_3$ observations, and how they are going to be used. $HNO_3$ is only mentioned in relation to other studies, see also my next point.

- At the end of the introduction, a guide through the manuscript connecting the parts to the objective raised in the abstract would be helpful, i.e. something like: "We use the $NO_2$ column data and anomalies correlated to Ap and QBO to assess the impact..." and "To identify another possible mechanism contributing to the stratospheric EPP-$NO_x$ variability, we evaluate MLS $HNO_3$ according to the QBO phase during the same period."

Observations and methods (Sect. 2)

- I couldn't find any methods presented here.

Sect. 2.1

- are the latitudes geographic or geomagnetic? I assume that the authors refer to geographic latitudes, for completeness, I suggest to state this somewhere in the (sub)section.

- l.97: I suggest to add some more details about the Aura satellite, such as orbit altitude, inclination, period, and local time.

- ll.102–104: I suggest to use: "*The latitudinal* coverage is illustrated... . *The figure*

shows ..."

- ll.105–107: This is a repetition of the earlier statement and can be removed.

- l.112: Noted in general comment 4, have the measurements been weighted according to their area when calculating the polar cap average (using cos(latitude) for example)? How do the authors account for the lack of measurements north of 70° during Aug–Sep? Has any correction been applied or is it implicitly assumed that the $NO_2$ column is constant (or zero) there? The latter is probably wrong, judging from the curved contours in Fig. 1.

- Table 1: Please indicate a range for all the values, for example using $\pm$ ($2\times$) standard error of the mean as in Fig. 4 or appropriate quantiles.

Sect. 2.2

- This section appears seemingly without relevance (see comment above about the introduction). It only becomes clear later in Sect. 4.3 when the authors discuss the possible influence of denitrification due to the formation of PSCs. I suggest to better explain how the data are relevant to the study.

- l.116: Geographic or geomagnetic latitudes? I suggest to state that somewhere at the beginning.

Sect. 2.3

- Mentioned in general comment 3, Ap has a non-normal distribution, how do the authors deal with that?

- l.122: the reference should be probably to Funke et al., 2014a instead of b.

- l.132: How was the confidence interval estimated?

Sect. 2.4

- Strahan et al., 2015 use a different definition of QBO which results in a different division of eQBO and wQBO years compared to the one presented here. The authors should comment on that and how it would influence the results (see also below).

- ll.136–137: This sentence is confusing, "take" does not seem to be appropriate here, please rephrase.

- Fig. 2: Error bars and a $\hat{A}_p = 8.5$ line would be helpful to visualize the Ap ranges and the division into low and high Ap years (only needed if Sect. 3.1 is kept in the manuscript, see below).

Results (Sect. 3)

- A little guide through the results would be helpful, as in "We investigate anomalies to assess ...", "Then, polar averages are correlated in order to ...", "Latitudinal correlations are used to ..."; either at the beginning of Sect. 3 or at the beginning of the respective subsections.

Sect. 3.1

As mentioned in general comment 1, this section does not seem to play a role in the rest of the manuscript and raises a lot of questions. For example, I count only two years (2005 and 2012) for panel (a), five (2007, 2009, 2010, 2014, and 2017) for panel (b), and three each for (c) and (d). How robust are those means then? How does it vary with the choice of QBO definition? Strahan et al. 2015 list 2011 as eQBO, not wQBO, how does that affect the results? How robust are the results with respect to the Ap distribution? 2017 for example could also be a high-Ap year (it is close), how would

that change Fig. 3?

- l.142: "... the mean deducted ..." What mean? The mean as shown in Fig. 1? If yes, please refer to that figure.

- However, I suggest to remove that section entirely and to start the results with the scatter plots in Sect. 3.2. The split into high and low Ap is not used later, the authors then only divide into eQBO and wQBO years.

Sect. 3.2

- Fig. 4 caption: "*The* yellow line ..."

- l.151: Again, please indicate if the data have been weighted by the area. It is only needed once, though. And again, what about the missing data in Aug–Sep?

- l.153: How was the linear fit achieved? Were the data weighted by their uncertainties or not? What about uncertainties in $\hat{A}_p$? Please be more specific here, in particular since this is later related to Funke et al., 2014a.

- l.156: "... [not] fully encompass the entire polar region ..." How do the authors deal with it? Are the averages calculated only up to $70°$ in those cases? Is the missing area filled with a constant value or even with zeros? I suggest to clarify these points.

- l.162: "... have consistently lower $NO_2$ column values, especially in August–September." May this be the result of omitting higher latitudes or implicitly replacing them by a constant or even zero? What about the influence of area (cos(latitude)) weighting?

- ll.163–174: Related to my general comment 2, how do the authors define the EPP part of the measured $NO_2$ columns? Why is Fig. A1 put into a non-existing appendix and not included here? I suggest to move that figure here as Fig. 5. Why not use the same Ap weighting scheme as described in Funke et al., 2014a? Note that they used

that procedure for a reason and it would make the two studies really comparable on an absolute scale.

Sect. 3.3

- l.182: Again, what part of the OMI $NO_2$ column is **EPP**-$NO_x$ here?

- l.185: How was the significance determined? Similar in caption of Fig. 5.

- l.186: I suggest to replace "from Fig. 5" by "*shown in* Fig. 5".

Discussion (Sect. 4)

- l.192: I suggest to add an article "... presented in *the* previous sections ..."

- l.192 cntd.: "less significant" than what? Using the frequentist language as in the other parts of the manuscript, the results are either significant or not (according to the chosen significance level). Do the authors mean "less correlated" ($\rho$ is around zero)? Or: "[the correlations] ... are less clear/smaller/weaker"?

- l.195: I suggest to remove "the month of".

- ll.196–211: As suggested in general comment 5, the study could be based on the polar vortex averages instead of the polar cap mean. I also suggest to move this part to the results, not the discussion.

- l.205: What about the vortex shape variability in other months?

- ll.208-211: I couldn't make any sense of that rather convoluted sentence, I suggest to rephrase it to be clearer; "thus" seems to be the wrong word here.

- l.212: The word "now" seems to be misused, I suggest to use "*in our study*".

- ll.213–214: Leaving the complications with Ap aside, the implication is only valid if the authors have a particular model/mechanism in mind that "generates a proportional response". Without that model or mechanism, the results merely *suggest* this response. I recommend to soften the wording accordingly, or to present a clear mechanism that links cause and effect.

- Fig. 6: Is this the October OMI ozone average column using all years? Or just one example month? What about the year-to-year variability of the vortex shape?

Sect. 4.2

Since this is the "discussion" section, the influence of the different QBO definitions should be discussed. The decreased $N_2O$ concentrations were observed in the average eQBO according to their (Strahan et al., 2015) definition of QBO (which is different from the one used here). Similarly, the mechanism that connects $N_2O$ and $NO_2$ could be repeated to make clear why the Strahan et al., 2015 study is relevant here.

- l.216: I suggest to swap "the" and "that".

- l.218: I suggest to remove "clearly".

- l.220: I suggest to replace "more" by "*a larger fraction*".

Sect. 4.3

Fig. 8: The panels are missing the (a), (b), and (c) indicators to be consistent with the figure caption. Caption (b): "anomaly from the mean", I assume the 3-day mean as shown in panel (a) is subtracted, please clarify that.

- l.222: I don't understand this sentence, what is meant by "the affected transport"? I suggest to rephrase that sentence to be clearer, and to remove "obviously" from it.

- l.224: An article seems to be missing: "*A* colder polar vortex ..."

- l.225: "As discussed earlier", where? A reference to the relevant section would be helpful.

- l.226–228: This sentence is hard to understand, I suggest to rewrite it, for example using *Thus* or *Therefore* instead of "So".

- l.234: I suggest to use "*down* to $-1$ ppbv" and to remove "clearly".

- ll.235–236: If PSCs are really responsible for the loss of $HNO_3$ due to denitrification, have the authors considered additional observations of e.g. PSC fraction or temperatures during eQBO or wQBO that would support that mechanism? I suggest to include a short comment or reference.

Conclusions (Sect. 5)

- ll.244–248: I suggest to move that part or a some version of it to the discussion section as it summarizes the assumed mechanisms. It would also fit at the end of the introduction to help the reader to understand the purpose of the study.

- l.252: This is a confusing sentence, how does the ozone hole suddenly come into play?

- ll.256–259: This conclusion is stretching it a bit too far in my opinion. According to the presented study, the EPP-$NO_x$ (in form of $NO_2$) does not change with QBO phase. Instead, the background $NO_2$ changes due to source and sink changes. As a consequence, the fraction of EPP-$NO_x$ ($NO_2$) on the overall amount varies with QBO phase. The authors may consider rephrasing their last conclusion a bit, such that the larger EPP-$NO_x$ *fraction* may need to be considered when considering the net effect of $NO_2$ on ozone chemistry (resp. recovery).

References

- There are two Seppälä et al., 2007 references listed, they should be separated with (a) and (b). They are referenced in ll.34 and 76 at least, which is which?

---

## Author Comment (AC1) · 13 Mar 2020

Author responses to reviewer comments on paper #acp-2019-1035 "EPP-NOx in Antarctic springtime stratospheric column: Evidence from observations and influence of the QBO".

We would like to thank the reviewer for their comments. Our detailed responses are given below.

Reviewer #1 (General comments):

1. The title of the paper suggests that polar springtime EPP-NOx is influenced by the QBO, however, none of the suggested mechanisms results in a modulation of the EPP contribution. Specifically, the authors suggest that (i) the "amount of the primary NOx source, N2O, transported into the polar regions" is affected by the QBO, and (ii) the "QBO affects the temperature of the polar vortex and thus the amount of denitrification". (i) would affect only the background NOx concentration (produced by N2O oxidation) and not the EPP contribution. (ii) would represent a total NOy loss mechanism (independently whether produced by EPP or N2O) and hence would not alter the relative EPP-NOx contribution. In the sense a title like "Evidence for EPP and QBO modulations of the Antarctic NO2 springtime stratospheric column from OMI observations" would be more appropriate.

   **Reply:** We would like to thank the reviewer for pointing this out. We very much agree and have changed the title of the paper as suggested.

2. It is suggested that, during eQBO, there is a lack of N2O transported to the polar regions which, in turn, results in a more prominent EPP-NOx contribution and hence better correlation of the observed NO2 column with Ap. This hypothesis is based on Fig 1 of Strahan et al. (2015) indicating a polar springtime N2O depletion during eQBO around 400-600 K (corresponding to approximately 15-25 km) from MLS observations. However, NOy production by N2O oxidation occurs predominantly at higher altitudes (peaking around 30 km which corresponds to a potential temperature level of around 800K) where the MLS observation analysed by Strahan et al. show a N2O increase during eQBO from the equator to around 70S. It is thus more likely that the background NO2 column is enhanced rather than decreased during eQBO because of increased N2O oxidation in the subpolar regions. Note that this is also in consonance with the results shown in Figures 3 and 4.

   **Reply:** This is correct, our original interpretation of Figure 1 of Strahan et al. (2015) was looking at the wrong altitude range. As pointed out, this is also in line with our results in figures 3 and 4. We have revised all text on this aspect and a grateful for the reviewer on pointing out this.

3. It is further suggested that the "QBO affects the temperature of the polar vortex and thus the amount of denitrification", resulting in smaller NO2 losses and hence in- creased NO2 during eQBO. The authors base this explanation on MLS HNO3 observations, indicating an HNO3 increase during eQBO in the 100-10 hPa range. However, it is not clear whether this increase is caused by reduced HNO3 losses (due to a warmer vortex and hence reduced PSC formation) or due to increased productions (e.g. by increased N2O oxidation as mentioned above). In order to proof their "denitrification" hypothesis, the authors should demonstrate that the HNO3 enhancements during eQBO are linked to temperature increases and/or PSC occurrence. In this context it is worth to mention that the link of PSC coverage and QBO modulation of polar temperature via the Holton-Tan effects is still under debate (see, e.g, Section 4 of Strahan et al., 2015).

   **Reply:** We have added analysis of MLS temperature observations analogous to the HNO$_3$ observations. These are presented side by side in Figure 1 here and now included in the

manuscript. The temperature analysis suggests that the stratosphere is typically warmer in Aug-Sept in eQBO years. We have revised the text to include the new analysis of the temperature observations to support the HNO₃ analysis and have added information about the temperature observations to the Methods section. We have also included the suggested reference to Strahan et al. on the state of knowledge on this topic in general.

*To test whether QBO phase affects denitrification in the Antarctic stratosphere, we analysed*

[Figure]

Figure 1: Temperature (left) and HNO₃ (right) mean fields (a-b) for the study period and anomalies for eQBO (c-d) and wQBO (e-f) phases

*temperature and HNO₃ observations from MLS (see section 2.2). Figure 8 a) and b) show the mean temperature and HNO₃ respectively, each averaged over $60°S$ to $82°S$ for 2005-2017 over the late winter–early spring period, i.e. when the polar vortex is coldest and PSCs are forming. Panels c) and e) show the anomalies from the mean temperature, for eQBO and wQBO years respectively. Panels d) and f) present the anomaly from the mean HNO₃ mixing ratio, for eQBO and wQBO years respectively. The vertical pressure range of all panels is 100 hPa to 10 hPa which corresponds to an altitude range of approximately 17 km to 32 km. Figure 8 suggest that eQBO years tend to have more HNO₃ (up to 1 ppbv) and higher temperature (up to 4 K) throughout this period, while wQBO years show a consistently negative anomaly in*

*$HNO_3$ (down to $-1$ ppbv) and lower temperature (down to $-4$ K). Colder temperatures would likely lead to more PSC formation and thus more $HNO_3$ being removed from the stratosphere (more denitrification) in wQBO years (than in eQBO years). It should, however, be noted here that the link of PSC coverage and QBO modulation of polar temperature via the Holton-Tan effects is still under debate (see, e.g, Strahan et al., 2015).*

Specific comments:

Comment: l23-25: Strahan et al. have shown that the lower stratospheric N2O anomaly at 450 K in the Antarctic polar springtime vortex correlates with the surfzone anomaly at 650 K 12 months earlier, the latter being characterized by enhanced N2O during eQBO.

Reply: We have revised the context of the Strahan et al. work through out our manuscript based on the comments the reviewer has provided above. We now write here:
*Recent work by Strahan et al. (2015) has shown that the phase of the QBO influences the transport of $N_2O$ from the surfzone to the polar vortex with a lag of 12 months. Further, their results (Figure 1 of Strahan et al. (2015)) indicate that easterly phase of the QBO during June-July is also generally associated with positive $N_2O$ anomalies in the polar stratosphere between altitudes of $\sim$24-33 km in September, and opposite for westerly phase of the QBO. Notably for our study, these particular altitudes, at this time, are also affected by large scale transport of mesospheric air masses affected by energetic particle precipitation (Funke et al., 2014a).*

Comment: l27: strictly speaking it is HNO3 (not NOx) being removed by denitrification.

Reply: This has been amended
*. . . in a process known as denitrification which removes $NO_x$ when it is stored in the $HNO_3$ reservoir*

Comment: l63: the major SSW occurred in January 2004 (not December 2003).

Reply: We have revised this text and it now reads: *However, dynamical effects, driven by the following major sudden stratospheric warming (SSW), indicated that this $NO_x$ was unlikely to have originated from the SPEs. . .*

Comment: l85-86: This sentence is a repetition of what is stated in the preceding paragraph.

Reply: We have removed this as suggested.

Comment: l87 "...whether this IS detectable...."

Reply: We have corrected this as suggested.

Comment: l147: It is the combined EPP and QBO influence which leads to the most prominent differences between H-Ap/eQBO and L-Ap/wQBO years.

Reply: We have changed the text to reflect this: *the combined influence of QBO and $\hat{A}_p$*

Comment: l186: Figure 5 shows correlations, not NO2 column increases.

Reply: We have revised this sentence to: *The results from Figure 5 suggest that the $NO_2$ increases at high polar latitudes in September are due to increased EPP/geomagnetic activity, as strong correlations between $NO_2$ and $\hat{A}_p$ occur in all panels.*

Comment: l204-205: What about wQBO? Fig 7a suggests that correlations improve also for wQBO when considering vortex-only observations.

Reply: We have revised Figure 7 to include the latitude correlations for both eQBO and wQBO years. We have also revised the text accordingly:
*Figure 7 c also shows higher correlation with more instances of significance in wQBO years in October than in Figure 5 c though this is more variable than in eQBO years (which is consistent with Figure 5, that wQBO years show lower correlation).*

Comment: l206: Consider to add "(see Fig. 7b)"

Reply: We have clarified this as suggested, with the text now reading: *Similarly for the horizontal distribution of the correlations (see Figure 7 b)). . .*

Comment: l215-220: see general comment (2)

Reply: See response to general comment 2. We have revised this text to clarify how our are results are related to the N2O transport discussed by Strahan et al.

Comment: l223-236: see general comment (3)

Reply: We have revised the text and included analysis of MLS temperature as suggested.

Comment: l229: QBO direction → QBO phase

Reply: We have now corrected this.

Comment: l251: "average rate" implies a time dependence. "average Ap dependence" would be clearer.

Reply: We have revised this as suggested.

Comment: l257: Why should total EPP-NOx only be accounted for in eQBO years?

Reply: This is very true. In writing this we were focused on the larger significances found during eQBO but have now removed this from this text.

[Figure]

Figure 2: Revised Figure 7, a) now area weighted, b) as before, c) showing the results inside the vortex for wQBO.

---

## Author Comment (AC2) · 13 Mar 2020

Author responses to reviewer comments on paper #acp-2019-1035 "EPP-NOx in Antarctic springtime stratospheric column: Evidence from observations and influence of the QBO".

We would like to thank the reviewer for their comments. Our detailed responses are given below.

Reviewer #2 (General comments):

1. Parts of the paper seem to be disconnected and there does not seem to be a clear logical thread to guide the reader. Some of the sections seem to focus on describing figures without explaining why they are relevant to the study and how they relate to the other sections. In particular Sect. 3.1 (esp. Fig. 3) seems unrelated and not relevant to the rest of the paper.

   **Reply:** We have taken this comment and several of the following comments and added more guidance to the reader, including the later proposed modifications to the introduction (subsections, further information on this work) and result sections. We have also added more clarification on why section 3.1 was included.

2. There seems to be a mixture of terms throughout the manuscript, "NOx", "NOy", and "NO2" seem to be used interchangeably. For example the title states "EPP-NOx" whereas the study focuses on NO2 only. Although according to Brasseur and Solomon, 2005, NO2 makes up around 80% of NOx in the stratosphere, this fact should be noted. The EPP part is not defined at all, Funke et al. 2014a,b use tracer correlations and Randall et al. 2007 use CH4–NO2 correlations to identify *EPP*-NOy, how do the authors discriminate between EPP and non-EPP NO2? A clear definition of these terms and how the authors use them should be given in Sect. 2.

   **Reply:** This is a very valuable comment. We have changed the title so that it now clearly states that we focus on $NO_2$. We agree that the terminology is not clear: many of the works looking at EPP impact on atmospheric odd nitrogen focus on the $NO_x$ family, as these gases are mainly available from observations and the immediate increases in odd nitrogen are visible in $NO_x$. Randall et al. (see e.g. Randall et al. 2006) initially coined the term EPP Indirect Effect (EPP IE) to describe the impact of $NO_x$ produced in the mesosphere by EPP that then descended into the stratosphere. Funke et al. have used a wide range of observations and have been able to investigate the wider $NO_y$ family which includes not only $NO_x$ but its long term reservoirs. For the context of these previous studies and our work both are important, thus we have used both terms $NO_x$ and $NO_y$ as appropriate (e.g. if we refer to a study that analysed $NO_y$, we use that term). Both have been described in the text and we have now further clarified the use of them across the paper.

   Several works have used tracer correlations to extract solely EPP produced $NO_x$ (or $NO_y$). Other works have investigated the variability in odd nitrogen resulting in variability in EPP levels (usually proxied by $A_p$). Our investigation is focused on finding evidence of EPP contribution to column observations, and we unfortunately do not have appropriate tracer information from OMI. Thus we are unable to robustly use the tracer method. To overcome this we performed correlation analysis both for latitudinal coverage and polar average $NO_2$ observations to find evidence of $A_p$ driven variability in the Antarctic $NO_2$ column.

   We added the following explanation of the $NO_x$ partitioning to section 2.1 following description of the effective vertical range of the OMI stratospheric $NO_2$ column:
   *At these altitudes, $NO_2$ makes up about 80% of the total $NO_x$ during daytime (see Brasseur and Solomon, 2005, chapter 5.5). As this corresponds to a large fraction of the total $NO_x$, we take the OMI $NO_2$ column measurements to represent a reasonable proxy for the variation in total $NO_x$.*

We also added the following explanation to section 2.3:
*Several previous works have used tracer correlations to extract solely EPP produced $NO_x$ (or $NO_y$ when information on the $NO_x$ reservoirs is available) (see e.g. Randall et al., 2007; Funke et al., 2014). When tracer information is not available, other works have investigated the variability in odd nitrogen resulting in variability in EPP levels (usually proxied by $A_p$) (see e.g. Seppälä et al., 2007). Here, we focus on finding evidence of EPP contribution to column observations. As we do not have mesosphere-stratosphere descent tracer observations available from OMI, we are unable to use the tracer correlation methods. To overcome this we perform correlation analysis both for latitudinal coverage and polar average $NO_2$ observations to find evidence of $A_p$ driven variability in the Antarctic $NO_2$ column.*

3. Do the authors average the 3h Ap or the daily mean Ap? Although other studies use average Ap as well (e.g. Funke et al. 2014a), Ap does not follow a normal distribution and the authors should be aware of that when using the mean as an estimator. This non-normality manifests itself in a very skewed distribution and a large standard devition, particularly the 3h values. Has this been considered in the correlation analysis? I suggest that the authors check that the mean is a valid estimator for the distribution or cite a relevant publication. I also suggest to present the Ap values with error bars in Figs. 2, 4, 7, and A1 and Table 1 (probably based on appropriate quantiles).

   **Reply:** We use the daily mean Ap values for calculation of our averages. We have also tested calculating median values instead of means, and this leads to the same year-to-year variability shown in Figure 2 which gives confidence that the mean-approach is valid here. We have added error ranges to Table 1 and Figure 2, but we found that the other figures became "too busy" when more lines were added. We are aware that $A_p$ has a skewed distribution, however, most previous studies also use mean or median values and following the same approach makes our results more comparable.

4. Is the 60°–90° average area weighted? If it is, it should be stated somewhere, Sect. 2 seems the obvious place (l.112). If not, higher latitudes may be artificially amplified in the polar cap average column. And in that case a discussion would be needed to assess the possible differences when taking area weighting into account.

   **Reply:** Originally the 60°–90° averages were not area weighted. We took this advice and found that area weighting improved our results so we are extremely grateful for pointing this out! Including the area weighting (cos(latitude)), we revised Figures 4 and 7a) and added appropriate description of the weighting method to the text and figure captions. Using identical area weighting to Funke et al. 2014a in Figure A1 also made our results quantitatively more comparable to theirs.

[Figure]

Figure 1: Revised Figure 4, now including area weighting.

[Figure]

Figure 2: Revised Figure A1, now including area weighting.

[Figure]

Figure 3: Figure 7 extended through November, a) now area weighted, b) showing results up until the polar vortex break up, c) showing the results inside the vortex for wQBO until the vortex break up.

5. In Sect. 4.1 the authors discuss the possible impact of out-of-vortex air on reducing the correlation between Ap and the NO2 column in October. Why is this presented in the "Discussion" section and not the "Results" section? As the authors seem to have an indication about the actual vortex available to them, why isn't the whole study based on vortex averages instead of whole polar cap averages? That would remove the ambiguity of including non-EPP-NO2 from horizontal transport/mixing in the polar cap average.

**Reply:** As suggested, we have moved this section to the Results as recommended.

Ideally, we would indeed prefer to do the entire analysis for inside vortex air only. However, the method used for locating the polar vortex only works when the ozone hole is present, i.e. mid-September until mid-November. Because of OMI's observation method and not having sufficiently low ozone amounts in the vortex area before that time, we are not able to do this for the entire time period. We have included here a version of Figure 7 which shows the latitude-resolved correlations throughout November. This shows how the vortex breaks up at this time and thus why we are not able to use the inner polar vortex data for months other than October. We only show October for the monthly mean (panel a)) as it is the only full month that we can calculate for with this method. This panel is now area weighted by cos(latitude). The final version of Figure 7 shows both eQBO and wQBO for the horizontal correlation, but is still only throughout October for the reasons given above

We have added the following description about the limits of our method:
*Note that we are unable to use this method on months prior to October due to OMI's viewing method and the ozone column not being sufficiently low to detect a clear vortex edge.*

[Figure]

Figure 4: Revised Figure 7, a) now area weighted, b) as before, c) showing the results inside the vortex for wQBO.

Specific comments:

Comment: ll.9–11: Does it really contribute to NO2 or is it just the fraction that changes due to a varying background? I suggest to rephrase these sentences to be clearer, for example how is it linked to the ozone hole? What is cause and what is the effect? See also my other comments below.

Reply: We have revised the text of the abstract to clarify that this $NO_x$ is a significant contributor to the polar $NO_2$ column when the background changes due to the QBO are accounted for. We have also further indicated the relevance of the ozone hole to our findings, as $NO_x$ catalytically destroys ozone.

*Our results suggest that once the background effect of the QBO is accounted for, $NO_x$ produced by EPP significantly contributes to the stratospheric $NO_2$ column at the time and altitudes when the ozone hole is present in the Antarctic stratosphere. Based on our findings, and the known role of $NO_x$ as catalyst for ozone loss ...*

Comment: I believe the introduction would profit from some additional subsections, e.g.: ll.33–48 "EPP indirect effect"; ll.49–83 "Previous work"/"Earlier studies"; ll.84ff "This work"

Reply: We have divided the introduction text into subsections as suggested.

Comment: ll.85–86: This is a repetition and can be removed.

Reply: We removed the sentence as suggested.

Comment: l.87: A verb is missing: "... this is detectable ..."

Reply: This has now been corrected.

Comment: The authors do not mention in the introduction that they are going to use (MLS) HNO3 observations, and how they are going to be used. HNO3 is only mentioned in relation to other studies, see also my next point.

Reply: We have added this to the introduction as suggested

*QBO conditions influence the probability of PSC formation, and as a result EPP-$NO_x$ may be less/more likely to be removed by denitrification during the winter/early spring during easterly/westerly QBO years. To test for the latter, we will analyse $HNO_3$ and temperature observations from the Microwave Limb Sounder (MLS), also on-board Aura.*

Comment: At the end of the introduction, a guide through the manuscript connecting the parts to the objective raised in the abstract would be helpful, i.e. something like: "We use the NO2 column data and anomalies correlated to Ap and QBO to assess the impact..." and "To identify another possible mechanism contributing to the stratospheric EPP-NOx variability, we evaluate MLS HNO3 according to the QBO phase during the same period."

Reply: This has been amended in the new subsection **This Work** (as suggested above), where we now write:

*Here, we use stratospheric $NO_2$ column observations from the Ozone Monitoring Instrument (OMI) on-board the Aura satellite to investigate EPP as a source of $NO_x$ in the Antarctic later-winter - spring. We have a relatively long satellite period (2005-2017) in which to analyse how this $NO_x$ propagates in the following springtime, and whether this is detectable in the $NO_2$ column. We also analyse how the phase of the QBO affects the contribution of EPP-$NO_2$ to the total $NO_2$ column in springtime. The QBO influence would be likely due to a combination of two effects: 1) QBO phase influences transport of $N_2O$ to the polar region $strahan_et_al_2015 resulting in a increased background NO_x$ source at key times and key*

*altitudes during easterly QBO years (opposite for westerly QBO); 2) QBO conditions influence the probability of PSC formation, and as a result EPP-NO$_x$ may be less/more likely to be removed by denitrification during the winter/early spring during easterly/westerly QBO years. To test for the latter, we will analyse HNO$_3$ and temperature observations from the Microwave Limb Sounder (MLS), also on-board Aura.*

*Comment:* Observations and methods (Sect. 2): I couldn't find any methods presented here.

*Reply:* We included a paragraph on how the correlations are calculated as part of this section. However, as this is only one short paragraph, we have revised the section title to "Data sets" as this hopefully better captures the satellite observations, EPP proxy, QBO, and the correlation description.

*Comment:* Sect. 2.1: are the latitudes geographic or geomagnetic? I assume that the authors refer to geographic latitudes, for completeness, I suggest to state this somewhere in the (sub)section.

*Reply:* All latitudes are geographic. We have added the following clarification in this section:
Note that all latitudes henceforth are geographic latitudes.

*Comment:* l.97: I suggest to add some more details about the Aura satellite, such as orbit altitude, inclination, period, and local time.

*Reply:* We have added the following sentence with a reference to an Aura validation paper:
Aura is in a Sun-synchronous orbit in the "A-train" constellation (orbital altitude of 705 km, inclination of 98°, 16 day repeat cycle), with ascending node crossing the equator approximately at 1:45pm daily (Schoeberl et al., 2008). As a result, OMI measurements take place at the same locations each year.

*Comment:* ll.102–104: I suggest to use: "The latitudinal coverage is illustrated... . The figure shows ..."

*Reply:* We have revised this text as suggested.

*Comment:* ll.105–107: This is a repetition of the earlier statement and can be removed.

*Reply:* We have removed this text as suggested.

*Comment:* l.112: Noted in general comment 4, have the measurements been weighted according to their area when calculating the polar cap average (using cos(latitude) for example)? How do the authors account for the lack of measurements north of 70° during Aug–Sep? Has any correction been applied or is it implicitly assumed that the NO2 column is constant (or zero) there? The latter is probably wrong, judging from the curved contours in Fig. 1.

*Reply:* As addressed in general comment 4 above, we now include area weighting. We do not make any assumptions about the missing values, i.e. we do not assign a value (zero or otherwise) to missing data. We only average the data available.

*Comment:* Table 1: Please indicate a range for all the values, for example using $\pm(2\times)$ standard error of the mean as in Fig. 4 or appropriate quantiles.

*Reply:* Error range has been added to $\hat{A}_p$ as $2\times$ standard error of the mean.

*Comment:* Sect. 2.2: This section appears seemingly without relevance (see comment above about the introduction). It only becomes clear later in Sect. 4.3 when the authors discuss the possible influence of denitrification due to the formation of PSCs. I suggest to better explain how the data are relevant to the study.

*Reply:* We have revised the text of the introduction to better relate the use of MLS $HNO_3$ to the study.

QBO conditions influence the probability of PSC formation, and as a result EPP-$NO_x$ may be less/more likely to be removed by denitrification during the winter/early spring during easterly/westerly QBO years. To test for the latter, we will analyse $HNO_3$ and temperature observations from the Microwave Limb Sounder (MLS), also on-board Aura.

*Comment:* l.116: Geographic or geomagnetic latitudes? I suggest to state that somewhere at the beginning.

*Reply:* This has been addressed in an earlier comment on section 2.1.

*Comment:* Sect. 2.3: Mentioned in general comment 3, Ap has a non-normal distribution, how do the authors deal with that?

*Reply:* This has been addressed in general comment 3 above.

*Comment:* l.122: the reference should be probably to Funke et al., 2014a instead of b.

*Reply:* This is has been corrected and now reads Funke et al., 2014a.

*Comment:* l.132: How was the confidence interval estimated?

*Reply:* The Spearman rank correlation calculation using Matlab's statistics package returns the corresponding $p$-values for testing the null hypothesis. We determine statistical significance using these $p$-values. We now write in the text:
Statistical significance is here defined as correlations significant at $\geq 95\%$ (i.e. $p$-value of $\leq 0.05$).

*Comment:* Sect. 2.4: Strahan et al., 2015 use a different definition of QBO which results in a different division of eQBO and wQBO years compared to the one presented here. The authors should comment on that and how it would influence the results (see also below).

*Reply:* The applicability of Strahan et al. 2015 is now addressed in Sec 4.2. Although their designation of QBO phase differs slightly from ours, this will not largely affect their comparability, as our designations only differ for one year of a 10 year study.

*Comment:* ll.136–137: This sentence is confusing, "take" does not seem to be appropriate here, please rephrase.

*Reply:* This sentence has been rephrased to:
We designate years when the zonal mean zonal wind direction is easterly as easterly QBO.

*Comment:* Fig. 2: Error bars and a $\hat{A}_p = 8.5$ line would be helpful to visualize the Ap ranges and the division into low and high Ap years (only needed if Sect. 3.1 is kept in the manuscript, see below).

*Reply:* The value 8.5 was a typo in the previous version of the manuscript, 8.3 being the correct average $\hat{A}_p$. We have revised all instances of this. We have also revised the figure to include errorbars and a line as suggested. The caption has also been revised accordingly.

[Figure]

Figure 5: Revised Figure 2, now including horizontal line at $\hat{A}_p$=8.3 and error bars on the $\hat{A}_p$ values.

*Comment:* *Results (Sect. 3): A little guide through the results would be helpful, as in "We investigate anomalies to assess ...", "Then, polar averages are correlated in order to ...", "Latitudinal correlations are used to ..."; either at the beginning of Sect. 3 or at the beginning of the respective subsections.*

*Reply:* *A motivation for each figure has been included at the start of each subsection:*
- We first investigate the anomaly from the mean for each of the four different categories of this study, eQBO H-$\hat{A}_p$, eQBO L-$\hat{A}_p$, wQBO H-$\hat{A}_p$, and wQBO L-$\hat{A}_p$. This is to show how $NO_2$ column evolves in the springtime in the different conditions and to further justify the splitting of years based on QBO phase.
- This is to investigate whether there is a relationship $\hat{A}_p$ size and $NO_2$ column, and how this is affected by QBO phase.
- This shows how correlation between $\hat{A}_p$ and $NO_2$ column evolves over time and latitude, and different QBO phase.

*Comment:* *Sect. 3.1: As mentioned in general comment 1, this section does not seem to play a role in the rest of the manuscript and raises a lot of questions. For example, I count only two years (2005 and 2012) for panel (a), five (2007, 2009, 2010, 2014, and 2017) for panel (b), and three each for (c) and (d). How robust are those means then? How does it vary with the choice of QBO definition? Strahan et al. 2015 list 2011 as eQBO, not wQBO, how does that affect the results? How robust are the results with respect to the Ap distribution? 2017 for example could also be a high-Ap year (it is close), how would that change Fig. 3?*

*Reply:* *As stated above, the earlier value of 8.5 was a typo. Our results in panel a) were actually for 2005, 2012 and 2017, and b) were 2007, 2009, 2010 and 2014. This figure is included to give the reader background on the background behaviour of $NO_2$ under each of the different conditions. It is intended to provide further justification for the methods used in the study (i.e. that both $\hat{A}_p$ and QBO affect seasonal evolution of column $NO_2$).*

*Comment:* *l.142: "... the mean deducted ..." What mean? The mean as shown in Fig. 1? If yes, please refer to that figure.*

*Reply:* *Yes it is the mean in Figure 1, we have clarified the text accordingly:* Figure 3 presents the average anomaly, (i.e. the mean as in Figure 1 deducted).

*Comment:* *However, I suggest to remove that section entirely and to start the results with the scatter plots in Sect. 3.2. The split into high and low Ap is not used later, the authors then only divide into eQBO and wQBO years.*

*Reply:* *The justification for this figure has been clarified.*
We first investigate the anomaly from the mean for each of the four different categories of this study, eQBO H-$\hat{A}_p$, eQBO L-$\hat{A}_p$, wQBO H-$\hat{A}_p$, and wQBO L-$\hat{A}_p$. This is to show how $NO_2$ column evolves in the springtime in the different conditions and to further justify the splitting of years based on QBO phase.

*Comment:* *Sect. 3.2: Fig. 4 caption: "The yellow line ..."*

*Reply:* *This has been added.*

*Comment:* *l.151: Again, please indicate if the data have been weighted by the area. It is only needed once, though. And again, what about the missing data in Aug–Sep?*

*Reply:* *This comment has been addressed above.*

*Comment: l.153: How was the linear fit achieved? Were the data weighted by their uncertainties or not? What about uncertainties in $\hat{A}_p$? Please be more specific here, in particular since this is later related to Funke et al., 2014a.*

*Reply: We use least squares linear fitting. This was mentioned in section 3.2 but we have added the information to the figure captions as well. We do not apply weighting as the errors in the $NO_2$ columns remain fairly consistent from year to year (see Figure 1 here). Note that the means are based on around $2 \times 10^5$ daily observations for each monthly mean.*

*Comment: l.156: "... [not] fully encompass the entire polar region ..." How do the authors deal with it? Are the averages calculated only up to $70°$ in those cases? Is the missing area filled with a constant value or even with zeros? I suggest to clarify these points.*

*Reply: As explained above, the averages are calculated for the actual data available. The missing area is treated as missing values and thus does not contribute to the mean. This has been further clarified in the text:*
The missing data in August and September is treated as missing values in the mean calculation and as such does not contribute to the mean in these figures.

*Comment: l.162: "... have consistently lower NO2 column values, especially in August–September." May this be the result of omitting higher latitudes or implicitly replacing them by a constant or even zero? What about the influence of area (cos(latitude)) weighting?*

*Reply: As stated earlier we did not apply area weighting (but do now, thank you for the suggestion). We do not assign values to missing values but rather omit them from calculations.*

*Comment: ll.163–174: Related to my general comment 2, how do the authors define the EPP part of the measured NO2 columns? Why is Fig. A1 put into a non-existing appendix and not included here? I suggest to move that figure here as Fig. 5. Why not use the same Ap weighting scheme as described in Funke et al., 2014a? Note that they used that procedure for a reason and it would make the two studies really comparable on an absolute scale.*

*Reply: Some of the aspect of this comment have been addressed in the earlier responses. The Ap weighting factors of Funke et al., 2014a are based on their linear regression of MIPAS EPP-$NO_y$ data and thus, unfortunately, not possible for us reproduce - we do not have tracer data or vertically resolved $NO_y$ needed for their sophisticated approach. Since the simple average $A_p$ approach has been found a good proxy for EPP by many previous studies, we rely on that here as well. With figure A1 we are concerned that it is too similar to figure 4 to be included in the main text. This was also advise from more senior colleagues. If moving the figure is supported by the ACP handling editor we would be very happy to move the figure to the main text.*

*Comment: Sect. 3.3: l.182: Again, what part of the OMI NO2 column is EPP-NOx here?*

*Reply: We have added information of the $NO_2$ contribution to the total $NO_x$ budget at the OMI effective altitudes (as discussed above). Since $NO_2$ makes up about 80 % of the total $NO_x$ and we find significant evidence of high positive correlation between this $NO_2$ column and $A_p$ this indicates that the EPP-$NO_x$ reported by others in past studies does indeed show up in the spring time column. We have added various clarifications to several parts of the paper that we think now clarifies these points more.*

*Comment: l.185: How was the significance determined? Similar in caption of Fig. 5.*

*Reply: We have added more information to earlier section 2 and this text has been clarified:*
Stippling indicates correlation significant at $\geq$95% level.

*Comment:* *l.186: I suggest to replace "from Fig. 5" by "shown in Fig. 5".*

*Reply:* *This has been revised as suggested*

*Comment:* *Discussion (Sect. 4): l.192: I suggest to add an article "... presented in the previous sections ..."*

*Reply:* *This has been revised.*

*Comment:* *l.192 cntd.: "less significant" than what? Using the frequentist language as in the other parts of the manuscript, the results are either significant or not (according to the chosen significance level). Do the authors mean "less correlated" ($\rho$ is around zero)? Or: "[the correlations] ... are less clear/smaller/weaker"?*

*Reply:* *We have revised this to:*
were found to have fewer occurrences of statistical significance in October than the surrounding months.

*Comment:* *l.195: I suggest to remove "the month of".*

*Reply:* *This has been removed as suggested.*

*Comment:* *ll.196–211: As suggested in general comment 5, the study could be based on the polar vortex averages instead of the polar cap mean. I also suggest to move this part to the results, not the discussion.*

*Reply:* *This has been moved to the results section as suggested. We cannot base this study on polar vortex averages as our method for isolating the vortex is not valid for the entire winter-spring period, see response to general comment 5.*

*Comment:* *l.205: What about the vortex shape variability in other months?*

*Reply:* *See response to general comment 5 on why the only full month this method is applicable for is October.*

*Comment:* *ll.208-211: I couldn't make any sense of that rather convoluted sentence, I suggest to rephrase it to be clearer; "thus" seems to be the wrong word here.*

*Reply:* *This sentence has been revised to:*
The reappearance of correlations in eQBO years in November in Figure 5b) is likely a mixing effect, with the break down of the polar vortex around this time leading to vortex air being mixed with extra-vortex air resulting in the $NO_2$ distribution not being as skewed as it was when contained in the vortex.

*Comment:* *l.212: The word "now" seems to be misused, I suggest to use "in our study".*

*Reply:* *We have revised this to:* has generally been lower in the past decade *to be more specific about the time period.*

*Comment:* *ll.213–214: Leaving the complications with Ap aside, the implication is only valid if the authors have a particular model/mechanism in mind that "generates a proportional response". Without that model or mechanism, the results merely suggest this response. I recommend to soften the wording accordingly, or to present a clear mechanism that links cause and effect.*

*Reply:* *We have removed this sentence.*

*Comment:* *Fig. 6: Is this the October OMI ozone average column using all years? Or just one example month? What about the year-to-year variability of the vortex shape?*

*Reply:* *We use co-located ozone observations for each individual NO$_2$ observation to determine the location of the polar vortex edge. Figure 6 is an example of one day, 19th October 2014. This is repeated every day in October in the study period to find the daily polar vortex extent. We then apply this to NO$_2$ to isolate the in-vortex NO$_2$ every day. We have revised both the text and the figure caption to make this clear.*

We perform this method for every day of October in the study period to find the daily vortex extent. This is then used to locate NO$_2$ that is inside the vortex for every day in October over the study period. An example of how the ozone column and the estimated vortex edge are reflected on the NO$_2$ column measurement for one day of the study (19th October 2014) is shown in Figure 6.

*The caption now reads:*

Vortex edge identification based on OMI ozone column for the 19th October 2014 . . . This method is repeated for every day in October throughout the study period.

*Comment:* *Sect. 4.2: Since this is the "discussion" section, the influence of the different QBO definitions should be discussed. The decreased N2O concentrations were observed in the average eQBO according to their (Strahan et al., 2015) definition of QBO (which is different from the one used here). Similarly, the mechanism that connects N2O and NO2 could be repeated to make clear why the Strahan et al., 2015 study is relevant here.*

*Reply:* *We have revised the text to address the differences in QBO defintion:* Although their designation of QBO phase differs slightly from ours, this will not largely affect their comparability, as our designations only differ for one year of a 10 year study. *We have repeated the mechanism connecting N2O and NO2 and clarified the relevance of the Strahan et al. study.*

*Comment:* *l.216: I suggest to swap "the" and "that".*

*Reply:* *This has now been corrected.*

*Comment:* *l.218: I suggest to remove "clearly".*

*Reply:* *We have revised this as suggested.*

*Comment:* *l.220: I suggest to replace "more" by "a larger fraction".*

*Reply:* *We have revised this as suggested.*

*Comment:* *Sect. 4.3: Fig. 8: The panels are missing the (a), (b), and (c) indicators to be consistent with the figure caption. Caption (b): "anomaly from the mean", I assume the 3-day mean as shown in panel (a) is subtracted, please clarify that.*

*Reply:* *This figure has been revised, with appropriate labels included. We have also updated the caption.*

*Comment:* *l.222: I don't understand this sentence, what is meant by "the affected transport"? I suggest to rephrase that sentence to be clearer, and to remove "obviously" from it.*

*Reply:* *We have revised this sentence to clarify the motivation behind this part of the discussion.* Here we discuss reasons for the consistently lower amounts of NO$_2$ in wQBO years that we pointed out in Figure 4. We suggest that this is due to the effect of denitrification in the polar region.

*Comment: l.224: An article seems to be missing: "A colder polar vortex ..."*

*Reply: We have now corrected this.*

*Comment: l.225: "As discussed earlier", where? A reference to the relevant section would be helpful.*

*Reply: We have revised the text to:*
As discussed in the Introduction, PSCs affect the heterogeneous chemistry in the polar region. . .

*Comment: l.226–228: This sentence is hard to understand, I suggest to rewrite it, for example using Thus or Therefore instead of "So".*

*Reply: We rewrote the sentence and it now reads:*
Thus, for years with more PSCs (i.e. wQBO) more denitrification would likely occur, resulting in the depleted $NO_2$ column reported here. This could also explaining the lower incidence of significant correlation in the wQBO cases.

*Comment: l.234: I suggest to use "down to $-1$ ppbv" and to remove "clearly".*

*Reply: We have revised the text as suggested.*

*Comment: ll.235–236: If PSCs are really responsible for the loss of HNO3 due to denitrification, have the authors considered additional observations of e.g. PSC fraction or temperatures during eQBO or wQBO that would support that mechanism? I suggest to include a short comment or reference.*

*Reply: We have revised this section to include analysis of MLS temperature observations in the polar lower stratosphere in the springtime, comparing the different QBO phases. The vortex is colder in wQBO years, supporting our hypothesis that PSCs are more likely to occur in wQBO years.*

*Comment: Conclusions (Sect. 5): ll.244–248: I suggest to move that part or a some version of it to the discussion section as it summarizes the assumed mechanisms. It would also fit at the end of the introduction to help the reader to understand the purpose of the study.*

*Reply: We took the advice and moved this section to the introduction as suggested. The end of the Introduction now reads:*
The QBO influence could be likely due to a combination of two effects: 1) QBO phase influences transport of $N_2O$ to the polar region strahan$_etal_2015resultinginaincreasedbackgroundNO_x$ source at key times and key altitudes during easterly QBO years (opposite for westerly QBO); 2) QBO conditions influence the probability of PSC formation, and as a result EPP-$NO_x$ may be less/more likely to be removed by denitrification during the winter/early spring during easterly/westerly QBO years. To test for the latter, we will analyse $HNO_3$ and temperature observations from the Microwave Limb Sounder (MLS), also on-board Aura.

Comment: l.252: This is a confusing sentence, how does the ozone hole suddenly come into play?

Reply: We agree with the reviewer. Our aim was to highlight the potentially changing role of solar activity via EPP as a modulation source of polar $NO_x$ as the stratospheric chlorine loading is changing. We have revised the text and it now reads:
We present evidence of contribution from EPP-$NO_x$ in the Antarctic stratosphere at a time when *halogen activate ozone loss is taking place*.

[Figure]

Figure 6: Temperature (left) and HNO$_3$ (right) mean fields (a-b) for the study period and anomalies for eQBO (c-d) and wQBO (e-f) phases

Comment: ll.256–259: This conclusion is stretching it a bit too far in my opinion. According to the presented study, the EPP-NOx (in form of NO2) does not change with QBO phase. Instead, the background NO2 changes due to source and sink changes. As a consequence, the fraction of EPP-NOx (NO2) on the overall amount varies with QBO phase. The authors may consider rephrasing their last conclusion a bit, such that the larger EPP-NOx fraction may need to be considered when considering the net effect of NO2 on ozone chemistry (resp. recovery).

Reply: We have revised this part according to the suggestion and it now reads:
*Our results suggest that, as chlorine activation continues to decrease in the Antarctic strato-sphere following the Montreal Protocol (Solomon et al., 2016), the total EPP-NO$_x$ (in addi-tion to SPEs as pointed out by Stone et al., 2018) should be accounted for in predictions of Antarctic springtime ozone recovery. Future studies should investigate the effects the larger EPP-NO$_x$ fraction when investigating the net effect of NO$_2$ on the fragile ozone chemistry in the springtime.*

Comment: References: There are two Seppälä et al., 2007 references listed, they should be separated

with (a) and (b). They are referenced in ll.34 and 76 at least, which is which?

Reply: This was an unfortunate issue with LaTeX and BiBTeX, which we used for compiling the bibliography. This has now been corrected with "a" and "b" added as appropriate.

---

## Author Response (AR2)

Author responses to reviewer comments on paper #acp-2019-1035 "Evidence for EPP and QBO modulations of the Antarctic $NO_2$ springtime stratospheric column from OMI observations".

We would like to thank the reviewer for their comments. Our detailed responses are given below.

Reviewer #1: The authors have addressed most of my comments satisfactorily, however I still have two concerns regarding the discussion of the QBO influence (section 4.1) and the influence of PSCs (section 4.2):

1. Regarding section 4.1, the sentence "Our results suggest that although the there is increased pool of N2O in polar region during eQBO contributing to larger NO2 column from N2O oxidation, EPP still contributes a clear fraction to the overall SH polar stratospheric column NO2 in the springtime" needs grammatical revision. Apart, while the revised interpretation provides an explanation for larger overall NO2 columns during QBOe years (independently on geomagnetic forcing) as indicated by Figures 3, 4, and 7a, it is not clear why the NO2-Ap correlation is more robust in QBOe years (see Figure 5), even when the analysis is restricted to the polar vortex (Figure 7 b and c). Although a deeper analysis of posible reasons might be beyond the scope of this papers, I think that this behavior should at least be mentioned.

   **Reply:** We have revised the text in section 4.1 accordingly and it now reads:
   *Our results suggest that although there is an increased pool of $N_2O$ in the polar region during eQBO (contributing to larger $NO_2$ column from $N_2O$ oxidation), EPP still contributes a clear fraction to the overall SH polar stratospheric column $NO_2$ in the springtime. Taking the QBO phase into account this EPP contribution during spring is generally more pronounced. We note that for wQBO years, the high correlation with $\hat{A}_p$ is only present until September, while lasting until November for eQBO years. It is not clear why this is the case for wQBO. Further analysis of possible reasons behind this discrepancy is beyond the scope of this work.*

2. Regarding section 4.2, to my opinion the analysis of MLS temperatures is not a stringent test whether the QBO phase affects denitrification, primarily because a lower mean temperature during QBOw does not necessarily imply a larger PSC area (this would be the case only if the area with temperature below the PSC threshold is larger in QBOw). In order to test the QBO impact on PSC formation one needs to look at the PSC areas as obtained e.g. from CALIPSO observations (see Fig. 13 of Pitts et al., 2018, www.atmos-chem-phys.net/18/10881/2018/, for the winters discussed here). Further, the bulk of PSC formation, particularly at altitudes above 20 km, occurs already in July-August, while the largest differences in MLS HNO3 are found in September (Fig 8b). Therefore I suggest to either extend section 4.2 by inclusion of additional analysis (including PSC observations) or to weaken related statements (i.e., l 5-9, l 100-102, and l 270 of the revised manuscript).

   **Reply:** We have modified the text to reflect the need for more detailed analysis for robust conclusions:

[revised manuscript text omitted]